# Multiplex image-based autophagy RNAi screening identifies SMCR8 as ULK1 kinase activity and gene expression regulator

Jennifer Jung[1], Arnab Nayak[1], Véronique Schaeffer[1], Tatjana Starzetz[2], Achim K Kirsch[3], Stefan Müller[1], Ivan Dikic[1,4,5], Michel Mittelbronn[2,6,7], Christian Behrends[1,8]*

[1]Institute of Biochemistry II, Goethe University School of Medicine, Frankfurt, Germany; [2]Neurological Institute, Goethe University, Frankfurt, Germany; [3]PerkinElmer, Inc., Hamburg, Germany; [4]Buchmann Institute for Molecular Life Sciences, Goethe University, Frankfurt, Germany; [5]Department of Immunology and Medical Genetics, School of Medicine, University of Split, Split, Croatia; [6]German Cancer Consortium, Heidelberg, Germany; [7]German Cancer Research Center, Heidelberg, Germany; [8]Munich Cluster for Systems Neurology, Ludwig-Maximilians-University Munich, Munich, Germany

**Abstract** Autophagy is an intracellular recycling and degradation pathway that depends on membrane trafficking. Rab GTPases are central for autophagy but their regulation especially through the activity of Rab GEFs remains largely elusive. We employed a RNAi screen simultaneously monitoring different populations of autophagosomes and identified 34 out of 186 Rab GTPase, GAP and GEF family members as potential autophagy regulators, amongst them SMCR8. SMCR8 uses overlapping binding regions to associate with C9ORF72 or with a C9ORF72-ULK1 kinase complex holo-assembly, which function in maturation and formation of autophagosomes, respectively. While focusing on the role of SMCR8 during autophagy initiation, we found that kinase activity and gene expression of ULK1 are increased upon SMCR8 depletion. The latter phenotype involved association of SMCR8 with the ULK1 gene locus. Global mRNA expression analysis revealed that SMCR8 regulates transcription of several other autophagy genes including WIPI2. Collectively, we established SMCR8 as multifaceted negative autophagy regulator.

*For correspondence: behrends@em.uni-frankfurt.de

## Introduction

Cellular integrity is dependent on vesicular transport in between membrane-bound compartments. Indispensable key players for intracellular trafficking are the Rab GTPases, which constitute the largest family of small Ras-like G-proteins. Rab GTPase cycling between the GDP- and GTP-bound, respective inactive and active states, is accelerated by Rab GTPase regulators. The intrinsic GTP hydrolysis of Rab GTPases is enhanced by Rab GAPs (GTPase activating protein), while Rab GEFs (guanine nucleotide exchange factor) induce the GTP-bound state. Active Rab GTPases recruit effector proteins that mediate vesicular budding, transport, targeting, tethering and fusion (see *Hutagalung and Novick (2011)* for review).

Autophagy is an intracellular quality and quantity control pathway during which diverse cytosolic cargoes such as damaged or surplus organelles, aggregated or misfolded proteins and pathogens are engulfed by double membrane structures coined autophagosomes and delivered for bulk

lysosomal degradation upon fusion of autophagosomes with lysosomes. This pathway originates from established membrane compartments such as the endoplasmic reticulum, where precursor structures and early autophagosome intermediates form. Phagophores and omegasomes are then elongated by vesicular input and give rise to autophagosomes upon their closure. At the molecular level, autophagy is initiated via inhibition of the mechanistic target of rapamycin complex 1 (mTORC1), causing formation of an active ULK1 (unc-51 like autophagy activating kinase 1) complex (*Kang et al., 2013*; *Kim et al., 2011*; *Jung et al., 2010*). The ULK1 complex comprises the kinase ULK1, FIP200/RB1CC1 (focal adhesion kinase interacting protein of 200 kDa/RB1 inducible coiled-coil1) (*Hara et al., 2008*), ATG13 (*Ganley et al., 2009*; *Hara et al., 2008*) and ATG101 (*Hosokawa et al., 2009b*; *Mercer et al., 2009*). Active ULK1 phosphorylates all of its complex partners as well as subunits of the PI3K (phosphatidylinositol 3-kinase) class III complex, namely Beclin1, Atg14L1 and hVps34 (*Jung et al., 2009*; *Russell et al., 2013*; *Egan et al., 2015*; *Park et al., 2016*). Phosphorylation of PI3K subunits induces generation of phosphatidylinositol-3-phosphate (PI3P) on the nascent phagophore (*Jaber et al., 2012*; *Burman and Ktistakis, 2010*), which subsequently recruits PI3P-binding proteins including WIPI2 (*Polson et al., 2010*). Consecutively, WIPI2 directs the ubiquitin-like conjugation machinery consisting of the ATG5~ATG12-ATG16L1 complex, ATG3, ATG7 and ATG10, to the elongated phagophore through association with ATG16L1 and hence, induces conjugation of mammalian ATG8s (LC3s and GABARAPs) to phosphatidylethanolamine (*Polson et al., 2010*; *Dooley et al., 2014*; *Proikas-Cezanne et al., 2004*; *Tanida et al., 2004*; *Weidberg et al., 2010*). Mammalian ATG8 proteins regulate cargo selection and autophagosome maturation through association with ATG8 family interaction motif (AIM, also known as LC3-interacting region (LIR)) containing proteins (*Behrends et al., 2010*; *Birgisdottir et al., 2013*) engaging separate, independent functions (*Weidberg et al., 2010*). While LC3s (LC3A, LC3B, LC3C) are required for phagophore elongation, GABARAPs (GABARAP, GABARAPL1, GABARAPL2) are indispensable for autophagosome maturation (*Weidberg et al., 2010*; *Joachim et al., 2015*). In addition, either group also operates in autophagy independent processes (*Stadel et al., 2015*; *Genau et al., 2015*). Finally, fusion of closed autophagosomes with lysosomes is promoted by recruitment of SNARE proteins such as STX17 and autophagosomal cargo is degraded (*Itakura et al., 2012*).

The involvement of certain Rab GTPases and GAPs in autophagy has already been analyzed in several unbiased mid- and large-scale screening studies (*Szatmári et al., 2014*; *Kern et al., 2015*; *Itoh et al., 2011*). For example, out of 36 TBC (Tre-2/Bub2/Cdc16) domain containing Rab GAPs 11 and 14 were found to inhibit starvation induced autophagy upon overexpression and to associate with human ATG8s, respectively, revealing TBC1D14 (*Lamb et al., 2016*; *Longatti et al., 2012*) and TBC1D5 (*Popovic et al., 2012*; *Popovic and Dikic, 2014*). Furthermore, several additional Rab GTPases and GAPs have been studied intensively including RAB7, TBC1D25, RAB33B, TBC1D2, RAB3GAP1 and RAB3GAP2 (*Itoh et al., 2011*; *Carroll et al., 2013*; *Spang et al., 2014*). However, many aspects of autophagy regulation in respect to membrane trafficking remain unclear. In particular, we know close to nothing about the function of Rab GEFs in autophagy.

To systematically identify autophagy-regulating Rab GTPases, GAPs and GEFs we performed an image-based RNAi screen monitoring a panel of early and late autophagosome markers in parallel at endogenous levels. Using this approach, we found and validated 34 candidates, of which seven (RAB27A (Ras-related protein Rab-27A), RAB27B, MADD (MAP kinase activating death domain), DENND2C (DENN domain containing 2C), RAB36, TBC1D8 (TBC1 domain family member 8) and SMCR8 (Smith-Magenis syndrome chromosomal region, candidate 8)) were selected for further characterization including electron microscopy and interaction proteomics. Very recently, several reports detected SMCR8 in complex with C9ORF72 and WDR41 (*Amick et al., 2016*; *Sellier et al., 2016*; *Sullivan et al., 2016*; *Yang et al., 2016*; *Xiao et al., 2016*; *Blokhuis et al., 2016*; *Ugolino et al., 2016*). This complex was further identified as RAB39B GEF, which promotes autophagic clearance of aggregated proteins (*Sellier et al., 2016*; *Yang et al., 2016*). Additionally, SMCR8 was implicated in mTORC1 regulation, lysosomal quality control and ULK1 modulation (*Amick et al., 2016*; *Sullivan et al., 2016*; *Sellier et al., 2016*; *Yang et al., 2016*; *Ugolino et al., 2016*). However, we provide evidence for the existence of a holo-assembly consisting of all ULK1 and SMCR8 complex subunits. Furthermore, SMCR8 depletion decreased phosphorylation of mTORC1 substrates but markedly enhanced ULK1 kinase activity. Unexpectedly, we found that SMCR8 repressed ULK1 gene expression independent of its GEF complex partners and regulated transcription of several other autophagy genes. Hence, we identified SMCR8 as versatile negative autophagy regulator.

## Results

### RNAi screen identifies autophagy-modulating Rab machinery components

Rab GTPases together with their activating (Rab GEFs) and inactivating (Rab GAPs) proteins are essential regulators of endomembrane trafficking. Since the involvement of these components in autophagy has not been systematically studied, we performed an unbiased, focused, image-based RNAi screen to identify Rab GTPases as well as their GEFs and GAPs that regulate autophagy. To comprehensively monitor the autophagy pathway at endogenous levels we first established parallel immunostaining in 384 well format for several autophagy markers (i.e. WIPI2, ATG12, LC3B, GABARAP and STX17), covering early autophagosome intermediates, autophagosomes and late autophagosomes. Using pooled siRNAs individually targeting each marker we confirmed antibody specificity in immunofluorescence (IF) and immunoblot analyses (*Figure 1—figure supplement 1A–E*). IF samples were measured on an automated confocal spinning disk microscope and spot numbers and their intensity were quantified and integrated using algorithm-based image analysis software. siRNA-mediated knockdown of Raptor or RAB7A significantly increased spot number and integrated spot signal (ISS) for all five markers while depletion of ATG12 or PIK3C3 significantly decreased the ISS across our marker panel (*Figure 1A and B*). Knockdown efficiency of these controls was confirmed by immunoblot or RT-qPCR analysis (*Figure 1—figure supplement 1F–H*). Screenability of our autophagy markers was assessed using the z'-factor, which evaluates the difference between the positive and negative control as well as the standard deviation. Importantly, z'-factors for all five markers were above 0.5 (*Figure 1B*), indicating excellent screening conditions.

Next, we performed the primary screen using 186 siRNA pools including all hitherto known or predicted human Rab GTPases, Rab GAPs and Rab GEFs (*Figure 1C*). Reverse siRNA transfection of U2OS cells was followed by fixation, parallel endogenous immunolabeling of all five established autophagosome markers and automated IF analysis. After normalization of spot numbers and ISS to non-targeting siRNA control (sicon), siRNA pools differing by more than two standard deviations for LC3B, GABARAP and STX17 or by more than three standard deviations for WIPI2 and ATG12 were selected as candidates. This primary screen yielded between 42 and 70 candidate siRNA pools per autophagosome marker that showed increased spot parameters while overall only up to 10 spot decreasing candidate siRNA pools were detected. Candidates of every autophagosome marker were ranked according to the maximal increase or decrease in spot numbers as well as ISS and the top ten altering siRNA pools for each individual marker plus those siRNA pools scoring for more than one marker were selected for deconvolution resulting in a total of 71 candidates (*Figure 1C*).

In the deconvolution screen cells were reversely transfected with four individual siRNAs per gene, fixed and immunolabeled for the respective autophagosome marker as before. After dataset normalization to sicon, a toxicity filter was applied excluding all siRNAs that caused broad variation in the number of cells or in the intensity or size of their nucleus or cytoplasm. To successfully pass deconvolution, candidates had to fulfill the standard deviation criterion applied above for three out of four individual siRNAs or for two out of three siRNAs in case one siRNA was removed due to cytotoxicity. In total, our deconvolution screen validated 34 candidate genes whose depletion resulted in an increase in spot numbers or ISS across our marker panel (11 for WIPI2, 15 for ATG12, four for LC3B, four for GABARAP and 15 for STX17) (*Figures 1C* and *2A*). Notably, none of the primary screen candidates whose knockdown decreased spot parameters passed our stringent deconvolution criteria. The detection of genes with known function in autophagy like RAB7A, RAB11B and several TRAPP components validated our screening results. As expected knockdown of RAB7A only increased spot parameters of late autophagy markers (LC3B, GABARAP and STX17). In addition, we identified several genes known to be involved in membrane trafficking such as TBC1D9B and RAB36 as well as completely enigmatic genes like DENND2C and TBC1D8 (*Figure 2A*).

To assess reproducibility, robustness and potential off-target effects of our screen we performed several quality control analyses. First, we examined biological replicates for LC3B and found significant correlation between normalized numbers of LC3B-positive spots in both experiments (*Figure 1D*), indicating high reproducibility. Second, the multiple correlation coefficient between single and pooled siRNAs across all autophagosome markers was calculated to be above the significance threshold of 0.3 (*Figure 1E*), suggesting valid candidate genes. Third, knockdown efficiency was determined for two siRNAs per validated candidate by measuring relative mRNA levels using

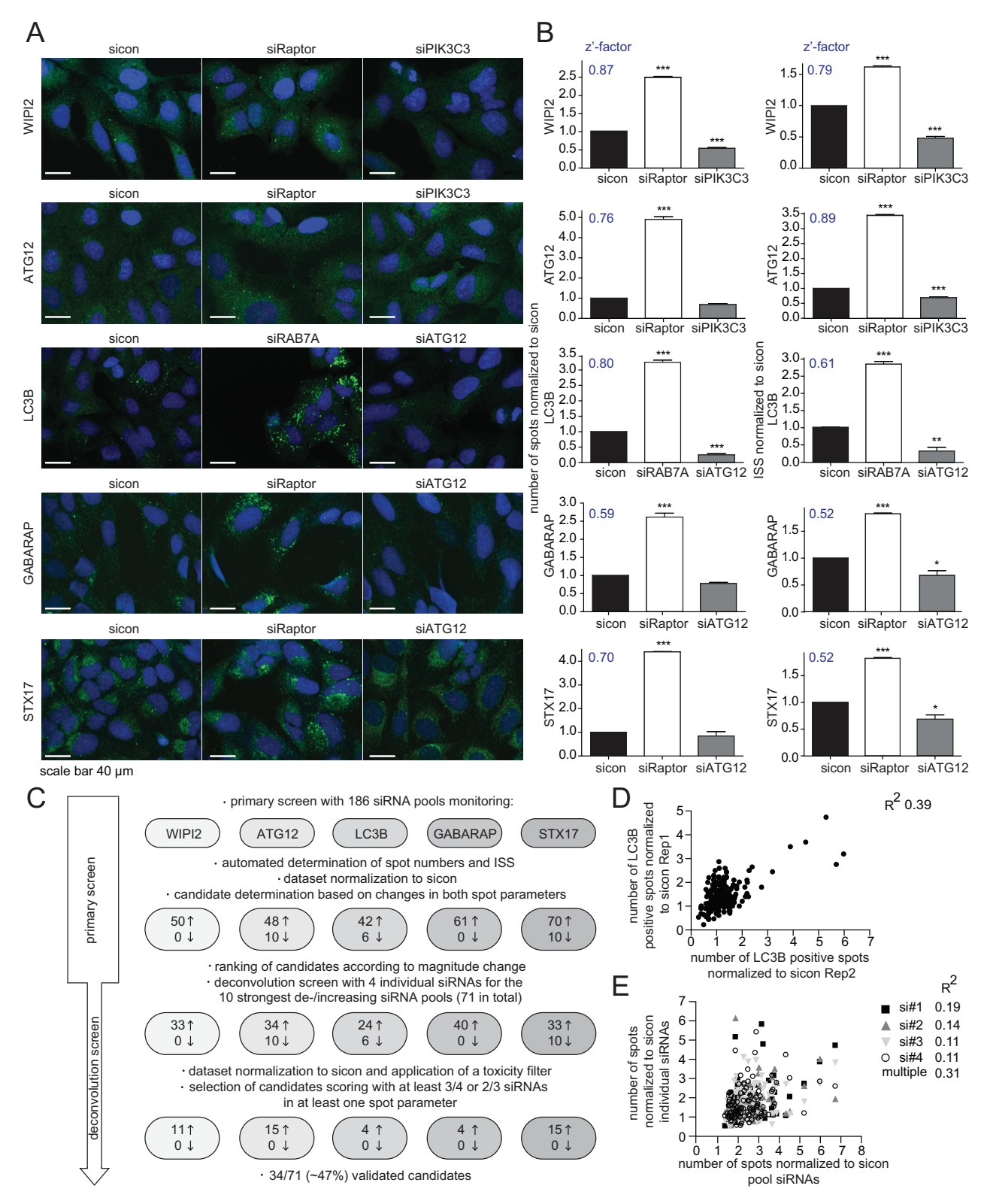

**Figure 1.** RNAi screen using endogenous autophagy markers identifies novel autophagy regulators among Rab GTPases and their regulators. (**A**) Parallel multiplex autophagosome monitoring. U2OS cells transfected for 72 hr with non-targeting control (sicon) or siRNA targeting known autophagy regulators, namely Raptor and RAB7A as positive controls and ATG12 and PIK3C3 as negative controls, were fixed and immunolabeled with anti-WIPI2, anti-ATG12, anti-LC3B, anti-GABARAP or anti-STX17 antibodies. Nuclei were counterstained with DRAQ5. Scale bars, 40 μm. (**B**) Automated

*Figure 1 continued on next page*

*Figure 1 continued*

quantification of number of spots and integrated spot signal (ISS) of at least 1000 cells per condition from images in (**A**). Error bars represent SEM. Significance was determined using one-way ANOVA compared with sicon. All experiments were performed n = 3. Calculated z'-factors are indicated for each antibody and for both spot parameters. (**C**) Overview of the screening strategy. Candidates that increase (arrow pointing up) or decrease (arrow pointing down) spot numbers and ISS in the primary and deconvolution screen are indicated. See *Figure 1—source data 1* and *Figure 1—source data 2* for complete results. (**D**) Correlation of number of LC3B-positive spots (normalized to sicon) from two biological replicates of the primary screen monitoring 186 siRNA pools for immunolabeled LC3B. $R^2$, Pearson's correlation coefficient. (**E**) Correlation of number of spots across all five autophagy markers (normalized to sicon) between pooled and individual siRNAs of candidates assayed in the deconvolution screen. $R^2$, Pearson's correlation coefficient.

The following source data and figure supplement are available for figure 1:

**Source data 1.** Primary image-based RNAi screen of 186 genes.
**Source data 2.** Deconvolution image-based RNAi screen of 71 genes.
**Figure supplement 1.** Evaluation of antibody specificity of early and late autophagosome markers.

RT-qPCR. 75% of all tested siRNAs showed a decreased mRNA level below 0.65 compared to sicon, while 25% had to be excluded due to potential off-target effects (*Figure 2B*, *Figure 2—figure supplement 1*).

## Treatment response, ultrastructural and interactome analysis of selected validated candidates

Based on knockdown efficiency, magnitude change in both spot parameters and literature curation, we selected seven candidate genes (DENND2C, MADD, RAB27A, RAB27B, RAB36, SMCR8 and TBC1D8) for further analysis. Knockdown with two individual siRNAs per candidate gene was performed in basal (DMSO), inducing (Torin1) and blocking (BafilomycinA1 (BafA1)) autophagy conditions prior to fixation, immunostaining and image analysis. While DENND2C, MADD, RAB27A, RAB27B, RAB36, SMCR8 or TBC1D8 depleted cells showed significantly increased spot numbers across several markers during basal autophagy as observed in our primary and deconvolution screens, depletion of either of these candidates led to a further increase in spot formation for at least one marker compared to sicon when cells were treated with Torin1 (*Figure 2C*, *Figure 2—figure supplement 2* and *Figure 2—figure supplement 3*). Importantly, re-examination of our marker panel under basal autophagy conditions with siRNAs from a different vendor largely confirmed the observed phenotypes across all seven candidates (*Figure 2—figure supplement 4A–C*).

All seven candidate genes were subjected to ultrastructural analysis. Electron microscopy revealed multi-lamellar bodies (*Hariri et al., 2000*) in RAB27A, RAB27B or MADD depleted cells and numerous vesicular structures with single or double-membranes upon TBC1D8, DENND2C or RAB36 knockdown (*Figure 3A*, *Figure 3—figure supplement 1A*). Importantly, both phenotypes were not observed in control cells. Moreover, in cells lacking SMCR8 an increased number of homogeneously electron-dense vesicles with varying diameters typically below 1 µm was observed, which potentially represented lysosomes (*Figure 3A*, *Figure 3—figure supplement 1A and B*).

Next, we generated stable 293T-REx cell lines inducibly expressing amino (N)-terminal hemagglutinin (HA)-tagged RAB27A, RAB27B, MADD, SMCR8, TBC1D8, RAB36 or DENND2C to determine the interactome of these candidates. Following cell lysis and HA-immunoprecipitation (IP), HA peptide eluted immune complexes were subjected to trypsin digestion, desalting and analysis by liquid chromatography tandem mass spectrometry (LC-MS/MS). High-confidence candidate interacting proteins (HCIPs) were identified by processing of mass spectral data using the CompPASS platform (*Behrends et al., 2010*; *Sowa et al., 2009*). Consistent with the role of RAB27A and RAB27B in melanosome transport (*Fukuda, 2013*), several components of this pathway (SYTL1, SYTL2, SYTL4, SYTL5, MYRIP and EXPH5) were among the HCIPs of both Rab27 proteins (*Figure 3B and C*). Further on, the autophagy regulators ATG2B (*Velikkakath et al., 2012*), SLC33A1 (*Pehar et al., 2012*), VMP1 (*Gilabert et al., 2013*; *Molejon et al., 2013*) and TM9SF1 (*He et al., 2009*) were detected as RAB27A HCIPs (*Figure 3C*, *Figure 3—figure supplement 2A*). In addition, RAB27B was associated

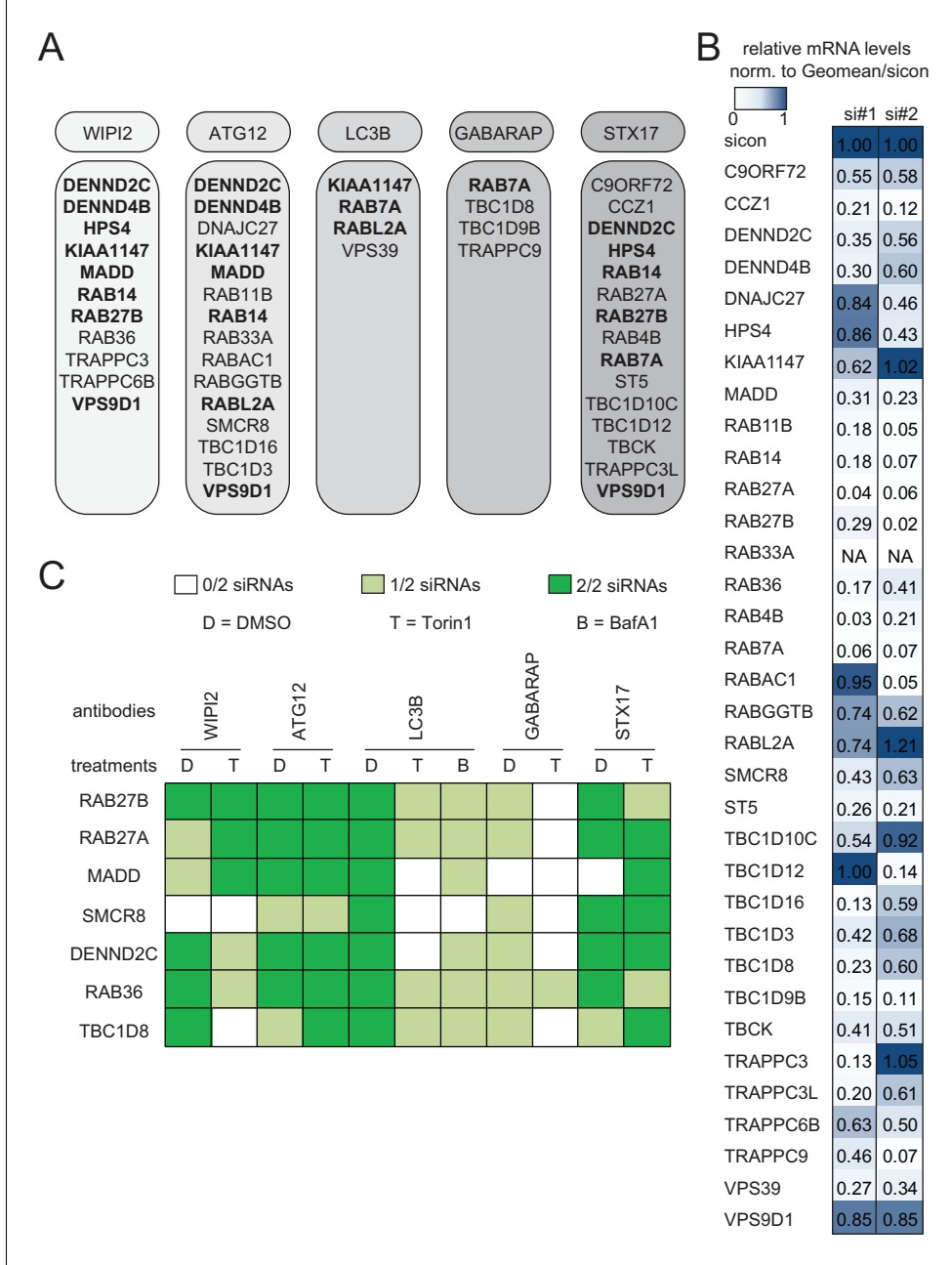

**Figure 2.** Knockdown efficiency and treatment response of validated candidates. (**A**) Overview of the validated candidates from the siRNA screen for each autophagy marker. Candidates that scored with more than one marker are indicated in bold. (**B**) Heatmap representation showing knockdown efficiency of two siRNAs for each of the 34 validated candidates that passed deconvolution. U2OS cells transfected with non-targeting control (sicon) or siRNAs targeting indicated candidates for 72 hr were harvested and subjected to mRNA isolation, reverse cDNA transcription and RT-qPCR with primers specific for the respective candidate gene. Relative mRNA levels were normalized to sicon. NA = Not analyzed. See *Figure 2—figure supplement 1* for complete results as well as *Supplementary file 1* and *Supplementary file 2* for siRNA and primer sequences. (**C**) Heatmap representation showing significant increase in WIPI2-, ATG12-, LC3B-, GABARAP- and STX17-positive spot numbers upon depletion of indicated candidates with one (light green) or two (dark green) oligos out of two siRNAs from (**B**). U2OS cells were transfected with non-targeting control (sicon) or siRNAs targeting indicated candidates for 72 hr and grown for 1 or 2 hr in the absence (D = DMSO) or presence of 250 nM Torin1 (T) or 100 nM BafilomycinA1 (BafA1, **B**), respectively. Following fixation cells were labeled with anti-WIPI2, anti-ATG12, anti-LC3B, anti-GABARAP or anti-STX17 antibodies and subjected to confocal microscopy. Number of spots were automatically

*Figure 2 continued on next page*

*Figure 2 continued*

quantified for at least 1000 cells per condition and normalized to sicon. See *Figure 2—figure supplement 2* for example images as well as *Figure 2—figure supplement 3* for complete results and statistics.

The following figure supplements are available for figure 2:

**Figure supplement 1.** Evaluation of knockdown efficiencies of deconvoluted candidates.

**Figure supplement 2.** Qualitative analysis of selected validated candidates upon autophagy stimulation.

**Figure supplement 3.** Quantitative analysis of selected validated candidates upon autophagy stimulation.

**Figure supplement 4.** Phenotype and knockdown evaluation of selected validated candidates by alternative siRNA oligos.

with its known GEF MADD (*Figure 3B*), which itself retrieved several kinases, amongst them the TFEB interactor PPP3CB (*Medina et al., 2015*) (*Figure 3D*). Importantly, IP of VMP1 and ATG2B with RAB27A was independently confirmed by immunoblotting (*Figure 3—figure supplement 2B*). Moreover, FYCO1 (*Pankiv and Johansen, 2010*; *Pankiv et al., 2010*) and the PIP2-binding protein ARHGAP26 (*Moreau et al., 2012*) were identified as HCIPs of DENND2C (*Figure 3E*), while RAB36 and TBC1D8 did not associate with known autophagy regulators (*Figure 3F and G*). Finally, the interactome of SMCR8 revealed members of the iron-sulfur cluster assembly machinery (FAM96B, CIAO1, GLRX5 and NUBPL) (*Stehling et al., 2012*), the lysosome maturation and trafficking BLOC-2 complex subunit HPS6 (*Bultema et al., 2012*), the Rab GEF C9ORF72 and its cofactor WDR41 (*Sellier et al., 2016*; *Xiao et al., 2016*; *Amick et al., 2016*; *Yang et al., 2016*; *Blokhuis et al., 2016*; *Ugolino et al., 2016*; *Zhang et al., 2012*; *Levine et al., 2013*) as well as the ULK1 complex component FIP200 (*Behrends et al., 2010*; *Sullivan et al., 2016*; *Sellier et al., 2016*; *Yang et al., 2016*) as prominent HCIPs (*Figure 3H*). Association of SMCR8 with the latter three was recently linked to autophagy modulation (*Sellier et al., 2016*; *Sullivan et al., 2016*; *Amick et al., 2016*; *Yang et al., 2016*). Since the role of Rab GEFs in autophagy regulation is largely unknown with the exception of the TRAPP complex (*Lamb et al., 2016*), we selected SMCR8 for further functional characterization.

## Differential binding of SMCR8 to the ULK1 complex components and C9ORF72

To validate autophagy-linked HCIPs within the SMCR8 network, we transiently expressed HA-tagged SMCR8 or ATG13, followed by HA-IP and immunoblotting. Indeed, the ULK1 complex members ULK1, FIP200 and ATG13 as well as C9ORF72 associated with tagged SMCR8 (*Figure 4A and B*), while endogenous SMCR8 was retrieved with HA-tagged ATG13 (*Figure 4C*). Importantly, the association of SMCR8 with ULK1 and FIP200 was confirmed at endogenous levels (*Figure 4D*). Next, we addressed whether the association of SMCR8 and C9ORF72 with each other or with the ULK1 complex is altered upon autophagy induction (*Figure 4E,F and G*). Consistent with recent work, starvation or Torin1 treatment did not affect the binding between SMCR8 and C9ORF72 (*Amick et al., 2016*; *Yang et al., 2016*). Intriguingly, IP of overexpressed HA-tagged or endogenous SMCR8 revealed an increased FIP200 binding to SMCR8 upon autophagy induction while SMCR8 interaction with ATG13 was reduced. Furthermore, C9ORF72 association with ULK1 complex components was remarkably sensitive to nutrient starvation as their interaction was almost undetectable in fed cells and increased substantially upon starvation. However, binding of SMCR8 to ULK1, FIP200 and ATG13 was more pronounced in fed cells compared to C9ORF72-ULK1 complex association in starved cells (*Figure 4F*). Together, these results indicate that binding of SMCR8 to C9ORF72 and the ULK1 complex is differentially regulated.

## SMCR8 binds ULK1 complex components and C9ORF72 via overlapping regions

To map the binding regions of ULK1 complex members and C9ORF72 on SMCR8 we employed a panel of cells transiently expressing HA-tagged full-length SMCR8 or fragments thereof followed by

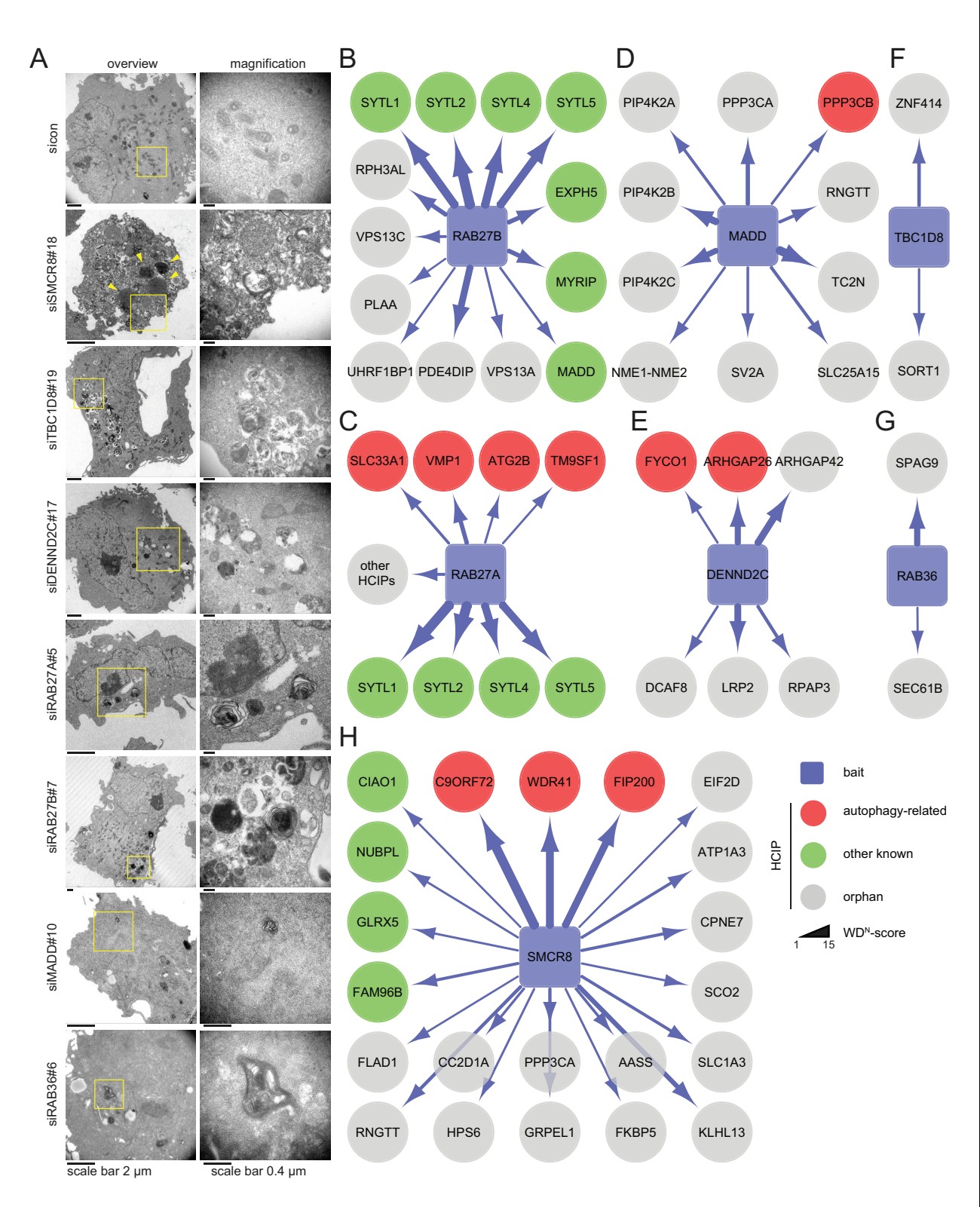

**Figure 3.** Ultrastructural analysis and interaction proteomics of selected validated candidates. (**A**) U2OS cells transfected with indicated siRNAs for 72 hr were harvested and subjected to sample preparation for electron microscopy followed by image acquisition. Scale bars, 2 and 0.4 μm as indicated. Arrowheads indicate homogeneously electron-dense vesicles. (**B–H**) Lysates of 293T-REx cells inducibly expressing indicated HA-tagged bait proteins (RAB27B (**B**), RAB27A (**C**), MADD (**D**), DENND2C (**E**), TBC1D8 (**F**), RAB36 (**G**) and SMCR8 (**H**)) were subjected to HA-IP, followed by trypsin digestion and

*Figure 3 continued on next page*

*Figure 3 continued*

mass spectrometric analysis. Individual interaction networks of indicated bait proteins with high-confidence candidate interacting proteins (HCIPs; average APSM of ≥2 and WDN score of ≥1) are color-coded according to autophagy-related (red), other known (green) and orphan (grey) association partners. Line thickness indicates interactions with WDN scores between 1 and 15. See *Figure 3—figure supplement 2* and *Figure 3—source data 1* for complete proteomic data.

The following source data and figure supplements are available for figure 3:

**Source data 1.** Complete interaction proteomics of 7 bait proteins.
**Figure supplement 1.** Phenotype and knockdown evaluation of selected validated candidates by alternative siRNA oligos for EM.
**Figure supplement 2.** Interaction network of RAB27A reveals interaction with ATG2B and VMP1.

HA-IP and immunoblot (*Figure 5A and B* and *Figure 5—figure supplement 1A and B*) or MS analysis (*Figure 5—figure supplement 1C and D*). Notably, SMCR8 fragments were designed in consideration of the domain boundaries of the tripartite DENN module, which is composed of the N-terminal u-DENN/longin, the central DENN, and the C-terminal d-DENN domains (*Zhang et al., 2012*) and bioinformatically predicted secondary structure elements (*Drozdetskiy et al., 2015*). This binding analysis revealed that the SMCR8 fragment spanning amino acids (aa) 120–320 was required and sufficient for the binding of ATG13 and immunoprecipitated even more endogenous ATG13 than full-length SMCR8. Conversely, association of FIP200 and ULK1 with SMCR8 was enhanced when the entire N-terminal fragment encompassing aa 1–700 was used and was dependent on the ATG13 binding site since further truncations of the N-terminus (compare fragments 271–700 and 1–700) reduced binding of FIP200 and ULK1 to SMCR8. As SMCR8 fragment 1–700 retrieved increased amounts of endogenous ULK1 and FIP200 compared to fragment 1–500, the region in between aa 500–700 of SMCR8 seemed particularly important for binding to ULK1 and FIP200. Furthermore, association of FIP200 and ULK1 with the SMCR8 fragment 1–700 was increased compared to full-length SMCR8 (*Figure 5A and B* and *Figure 5—figure supplement 1A–D*), indicating a potential inhibitory role of the C-terminal region of SMCR8 spanning aa 701–937, which itself did not interact with any of the tested binding partners (*Figure 5—figure supplement 1A and B*). For C9ORF72, a SMCR8 fragment consisting of aa 1–400 was necessary and sufficient to mediate binding. Further N-terminal or C-terminal clipping of the SMCR8 fragment 1–400 completely abolished binding of C9ORF72. Interestingly, SMCR8 fragment 1–400 showed strongly reduced interaction with ATG13 compared to fragment 1–320, suggesting that the region encompassing aa 320–400 of SMCR8 has an inhibitory and promoting role in the association with ATG13 and C9ORF72, respectively (*Figure 5A and B*). Since these results provide evidence for tight association of SMCR8 with ULK1 complex members and C9ORF72 via overlapping binding regions (*Figure 5C*), we examined whether ATG13 and C9ORF72 compete for binding to SMCR8. However, increasing amounts of exogenously expressed GFP-tagged ATG13 or C9ORF72 were not able to outcompete C9ORF72 or ATG13 from SMCR8 immune complexes (*Figure 5D and E*). Further on, SMCR8 overexpression or depletion did not alter association between ULK1 complex components (*Figure 5F and G*).

## SMCR8 is part of a C9ORF72 complex and a C9ORF72-ULK1 complex holo-assembly

To start addressing whether SMCR8 associates with its binding partners in two distinct complexes or in one holo-assembly, we subjected eluted immune complexes of exogenously expressed HA-tagged SMCR8, C9ORF72, ATG13 and ULK1 to Native PAGE followed by immunoblot or MS analysis (*Figure 6A*). Together with WDR41 but in the absence of any ULK1 complex component, SMCR8 and C9ORF72 formed a stable complex whose migration in Native PAGE peaked between 480 and 720 kDa. As reported *Mercer et al. (2009)*, ATG13 associated with ATG101 and formed a similar size complex that likewise lacked FIP200 and ULK1, the latter of which also existed unbound by its complex partners. However, all SMCR8-binding partners, namely C9ORF72, WDR41 as well as the ULK1 complex were also present in a second higher molecular weight assembly that migrated between 720 and 1200 kDa. Complementary size exclusion chromatography (SEC) experiments of

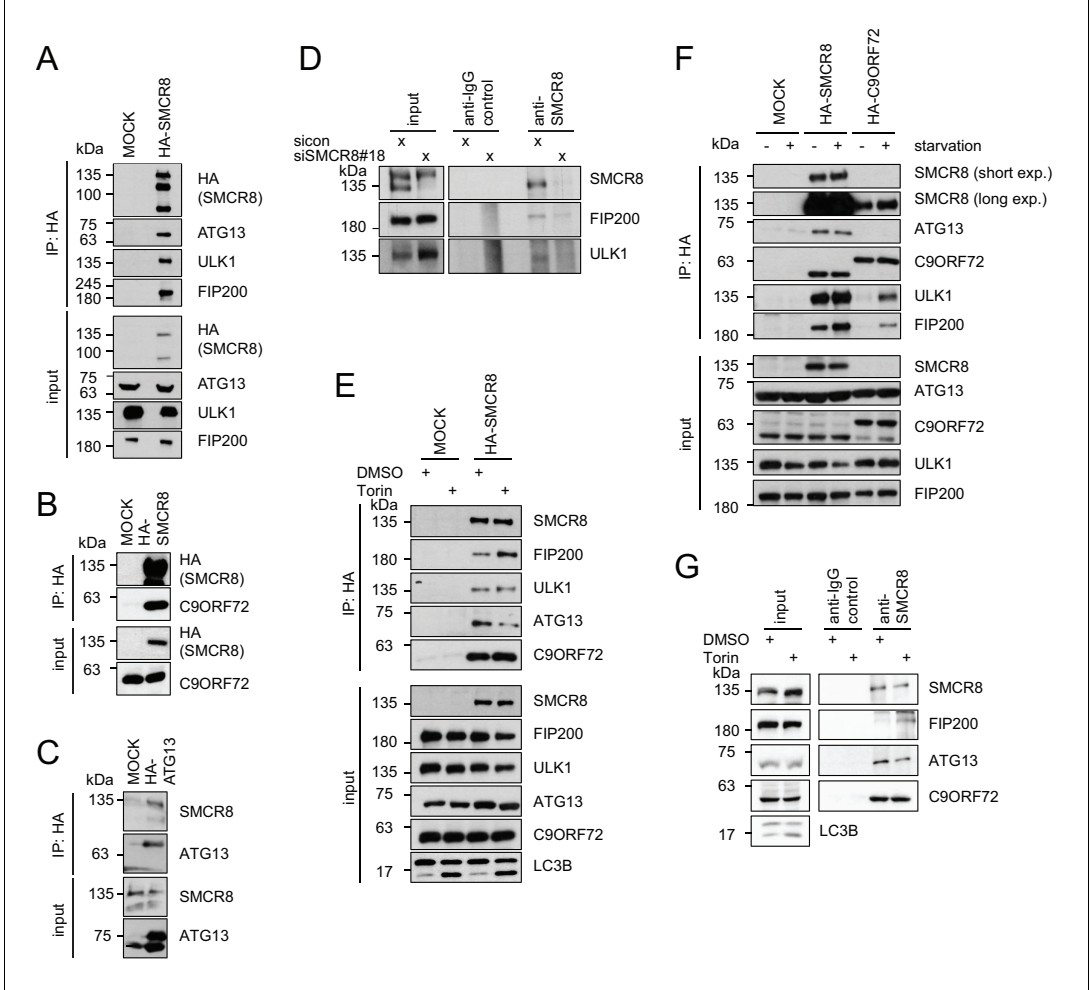

**Figure 4.** SMCR8 associates with ULK1 complex components and C9ORF72. (**A–C**) Empty 293T-REx cells (MOCK) or those inducibly expressing HA-tagged SMCR8 (**A,B**) or ATG13 (**C**) were lysed and subjected to HA-IP, followed by SDS-PAGE and immunoblotting with indicated antibodies. (**D**) 293 T cells transfected with non-targeting control (sicon) or SMCR8 siRNA for 72 hr were lysed, followed by IP with anti-SMCR8 or anti-IgG as control. Co-immunoprecipitated proteins were separated by SDS-PAGE and analyzed by immunoblotting. (**E**) Empty 293T-REx cells (MOCK) or those inducibly expressing HA-tagged SMCR8 were grown in the absence (DMSO) or presence of 250 nM Torin1 for 2 hr and analyzed as in (**A**). (**F**) Empty 293T-REx cells (MOCK) or those inducibly expressing HA-tagged SMCR8 or C9ORF72 were starved with EBSS for 2 hr and analyzed as in (**A**). exp. = exposure. (**G**) 293 T cells transfected with non-targeting control (sicon) or SMCR8 siRNA for 72 hr were grown in the absence (DMSO) or presence of 250 nM Torin1 for 2 hr, prior to lysis, followed by IP with anti-SMCR8 or anti-IgG as control. Co-immunoprecipitated proteins were separated and detected by SDS-PAGE and immunoblotting, respectively.

whole cell lysates followed by immunoblot analysis confirmed the distribution of the ULK1 complex in a high molecular weight assembly above 1 MDa (*Figure 6B*). Accordingly, SMCR8 was detected in fractions between 440 and 669 kDa in a SMCR8-C9ORF72-WDR41 complex and above 669 kDa in a SMCR8-C9ORF72-WDR41-ULK1 complex holo-assembly (*Figure 6B*). Notably, specificity of the anti-SMCR8 antibody was verified with SEC of whole cell lysates from SMCR8 knockdown cells (*Figure 6—figure supplement 1A*). In agreement with our co-immunoprecipitation experiments (*Figure 5G*), SMCR8 depletion left the ULK1 complex distribution unchanged (*Figure 6—figure supplement 1A*). To gain more insights into the SMCR8-C9ORF72-ULK1 holo-complex, we combined IP of HA-tagged ATG13 with SEC and MS analysis (*Figure 6C*). The size fractionation pattern of eluted ATG13 immunoprecipitates revealed three distinct populations of the common SMCR8-C9ORF72-WDR41-ULK1 complex assembly, which peaked at approximately 500 kDa, 1 MDa and several MDa, respectively, and might represent monomeric and multimeric states of this holo-assembly as suggested previously for the ULK1 complex (*Hosokawa et al., 2009a*; *Köfinger et al., 2015*). Since

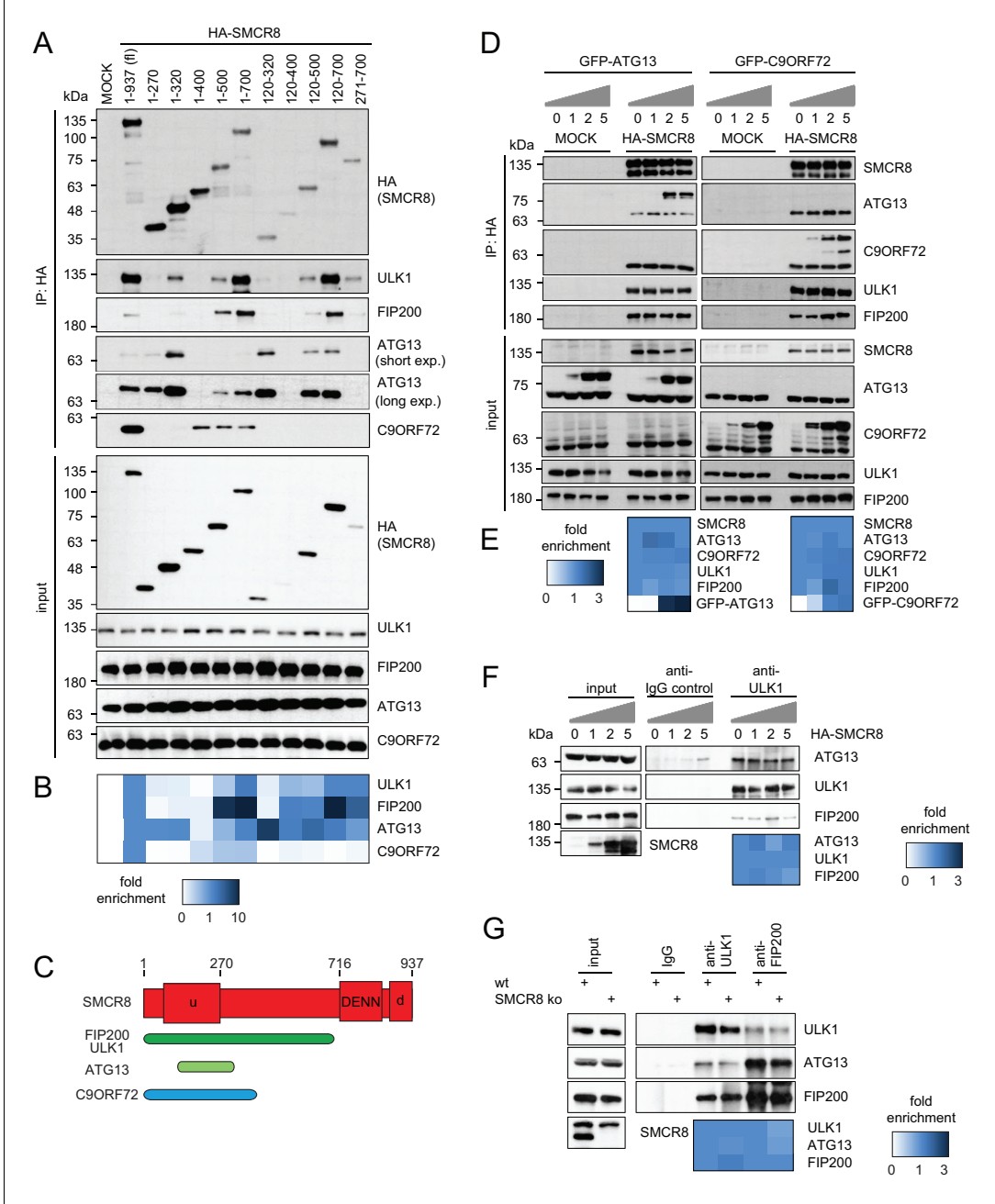

**Figure 5.** SMCR8 employs overlapping binding regions to associate with ULK1 complex components and C9ORF72. (**A**) 293 T cells transiently transfected with HA-tagged full length (fl) SMCR8 or indicated fragments thereof were lysed and subjected to HA-IP and analyzed by SDS-PAGE and immunoblotting with indicated antibodies. exp. = exposure. (**B**) Immunoblots in (**A**) were quantified using ImageJ. Co-immunoprecipitated proteins were normalized to the amount of HA-tagged SMCR8 fragments. Fold enrichment compared to full length SMCR8 was calculated and displayed as heatmap. (**C**) Domain architecture of SMCR8 with mapped binding regions for FIP200, ULK1, ATG13 and C9ORF72. (**D**) Empty 293T-REx cells (MOCK) or those inducibly expressing HA-tagged SMCR8 were transfected with GFP-ATG13 or GFP-C9ORF72, lysed and subjected to HA-IP, followed by SDS-PAGE and immunoblotting with indicated antibodies. (**E**) Immunoblots of 3 independent experiments in (**D**) were quantified using ImageJ. Co-immunoprecipitated proteins were normalized to the amount of HA-tagged SMCR8. Fold enrichment compared to SMCR8 was calculated and displayed as heatmap. (**F**) 293 T cells transfected with increasing amounts of HA-tagged SMCR8 were lysed, followed by IP with anti-ULK1 or anti-IgG as control. Co-immunoprecipitated proteins were analyzed as in (**D**). Immunoblots of 3 independent experiments were quantified using ImageJ. Co-immunoprecipitated proteins were normalized to the amount of ULK1. Fold enrichment compared to ULK1 was calculated and displayed as heatmap. (**G**) Lysates from 293 T SMCR8 wildtype (wt) or knockout (ko) cells were subjected to IP with anti-ULK1, anti-FIP200 or anti-IgG as control. Co-immunoprecipitated proteins were analyzed as in (**D**). Immunoblots of 3 independent experiments were quantified using ImageJ. Co-

*Figure 5 continued on next page*

Figure 5 continued

immunoprecipitated proteins were normalized to the amount of ULK1 or FIP200, respectively. Fold enrichment compared to ULK1 or FIP200 was calculated and displayed as heatmap.

The following figure supplement is available for figure 5:

**Figure supplement 1.** SMCR8 employs overlapping binding regions to associate with ULK1 complex components and C9ORF72.

autophagy induction resulted in enhanced interaction of SMCR8 and C9ORF72 with the ULK1 complex (*Figure 4F*), we examined the fractionation pattern of these components in whole cell lysates or eluted immune complexes by SEC upon Torin1 treatment. However, we could not observe major changes in the distribution of the ULK1 complex components, SMCR8 or C9ORF72 under these conditions (*Figure 6—figure supplement 1B–E*). In summary, our results indicate that SMCR8 binds C9ORF72 and WDR41 to form a stable complex that is joined by the ULK1 complex to form an even larger combined assembly.

## SMCR8 regulates both initiation and maturation of autophagosomes

Although Charlet-Berguerand and colleagues recently demonstrated GEF activity of SMCR8 in complex with C9ORF72 and WDR41 towards RAB8A and RAB39B of which the latter is required to promote clearance of aggregated proteins dependent on SMCR8 phosphorylation by TBK1 (*Sellier et al., 2016*), the exact function of SMCR8 in autophagy is far from being clearly understood. Given that SMCR8 knockdown increased spots of early and late autophagosome markers across our different screening efforts (*Figure 2A and C*, *Figure 2—figure supplement 4A–C*), we performed an additional series of experiments to unequivocally establish a role of SMCR8 in formation or maturation of autophagosomes. Briefly, we analyzed cells stably expressing RFP-GFP-LC3B through which autophagosomes can be distinguished from autolysosomes due to loss of pH-sensitive GFP fluorescence in autolysosomes (*Kimura et al., 2007*). Upon SMCR8 knockdown, we observed an increase in total number of spots (*Figure 6—figure supplement 2A and B*), which can result from enhanced formation of autophagosomes or from blockage of autophagosome maturation. Indicative of the latter is that the ratio of autophagosomes to autolysosomes in SMCR8 depleted cells (at least for siSMCR8#20) showed a considerable shift in favor of autophagosomes. Furthermore, treatment of control cells with BafA1 increased the ratio and the total number of spots as expected (*Figure 6—figure supplement 2A and B*). Additional loss of SMCR8 slightly aggrandized this increase, suggesting a role of SMCR8 in formation of autophagosomes. At last, Torin1 treated cells depleted of SMCR8 further accumulated spots, which is again an indication for blockage of autophagosome maturation. Together, we concluded that SMCR8 exerts two independent functions in autophagy. On one hand SMCR8 represses autophagosome formation and on the other hand SMCR8 promotes autophagosome maturation.

## SMCR8 depletion increases the formation of early autophagosome intermediates

Subsequently, we focused our efforts on addressing how SMCR8 affects autophagosome formation. Given the fact that SMCR8 is part of the ULK1 complex, we examined the localization of ULK1 and FIP200 in cells lacking SMCR8. In addition, we also monitored the PI3P effector WIPI2 as surrogate for hVps34 activity, which is regulated by ULK1 (*Russell et al., 2013*). Intriguingly, SMCR8 knockout (ko) cells displayed a significant increase in spot numbers of all three markers (*Figure 6—figure supplement 3A–D*). To test whether these spots indeed represent early autophagosome intermediates positive for both ULK1 and WIPI2, we monitored the subcellular distribution of both markers in SMCR8 depleted cells. Consistent with our results in SMCR8 knockout cells, loss of SMCR8 yielded significantly elevated numbers of ULK1 positive spots (*Figure 6—figure supplement 3E and F*). Conversely, but in agreement with our initial screening results, RNAi-mediated knockdown of SMCR8 was not sufficient to phenocopy the increase in WIPI2 positive spots observed in cells completely lacking SMCR8 (*Figure 2A*, *Figure 6—figure supplement 3E and F*). Importantly, colocalization of ULK1 with WIPI2 increased about 2-fold upon SMCR8 depletion in basal and Torin1-

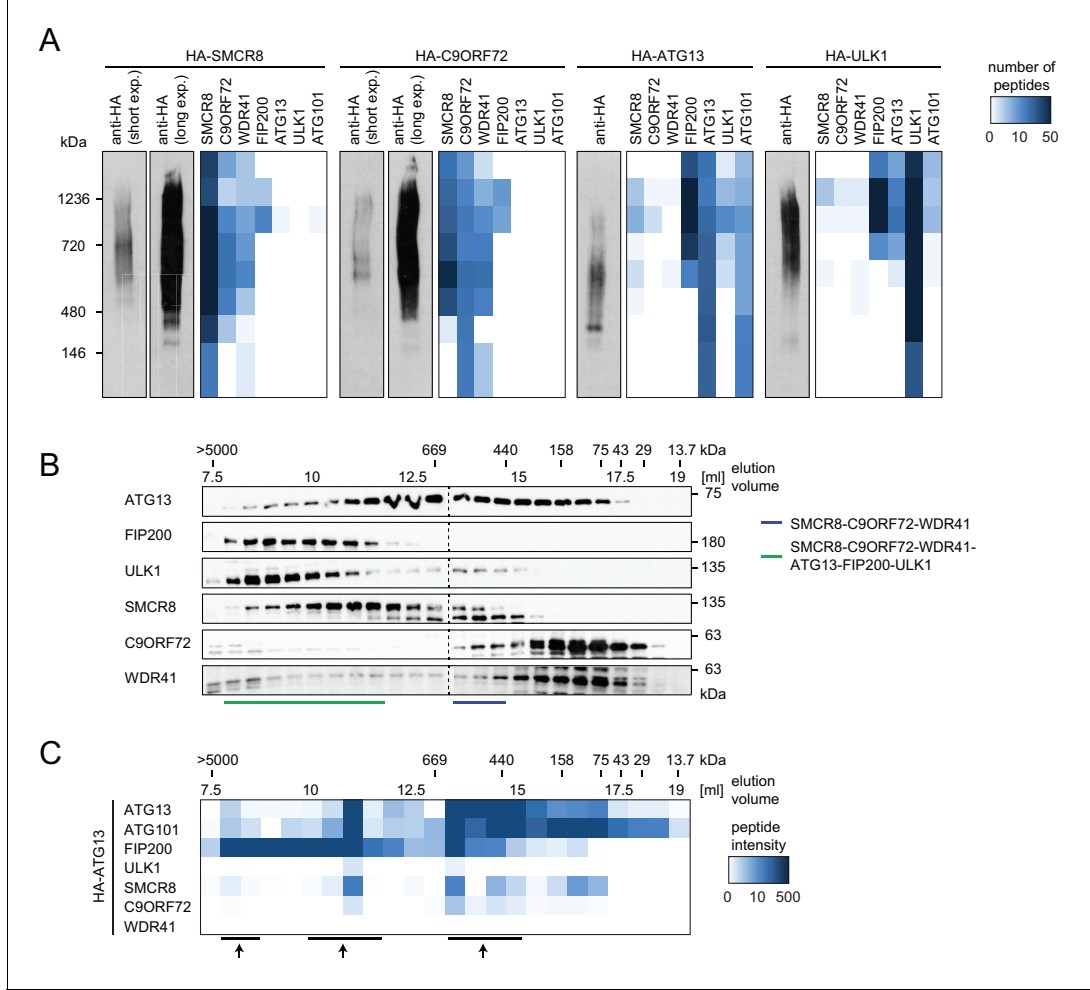

**Figure 6.** SMCR8 is part of a C9ORF72 complex and a C9ORF72-ULK1 complex holo-assembly. (**A**) 293T-REx cells inducibly expressing HA-tagged SMCR8, C9ORF72, ATG13 or ULK1 were lysed and subjected to HA-IP, followed by Native PAGE and immunoblotting or mass spectrometric analysis. Number of peptides is depicted as heatmap representation. exp. = exposure. See *Figure 6—source data 1* for complete proteomic data. (**B**) 293 T cells were lysed via freeze-thaw cycles and subjected to SEC, followed by SDS-PAGE and immunoblot with indicated antibodies. SMCR8 complexes are indicated with bars. (**C**) 293T-REx cells inducibly expressing HA-tagged ATG13 were lysed and subjected to HA-IP, followed by SEC and mass spectrometric analysis. Peptide intensity is depicted as heatmap representation. Arrows indicate intensity peaks. See *Figure 6—source data 2* for complete proteomic data.

The following source data and figure supplements are available for figure 6:

**Source data 1.** Proteomic data from Native PAGE analysis of HA-IPs.

**Source data 2.** Proteomic data from SEC of immunoprecipitated HA-ATG13.

**Figure supplement 1.** Composition of SMCR8-containing complexes is unchanged in response to Torin1 treatment.

**Figure supplement 2.** Dual role of SMCR8 in regulating initiation and maturation of autophagosomes.

**Figure supplement 3.** SMCR8 depletion induces formation and colocalization of ULK1- and WIPI2-positive structures.

treated conditions (*Figure 6—figure supplement 3E and G*), suggesting an enhanced formation of early autophagosome intermediates in the absence of SMCR8.

## SMCR8 regulates ULK1 kinase activity

To gain further insights into the mechanism of SMCR8-mediated autophagy initiation restriction, we examined the effect of SMCR8 depletion on ULK1 kinase activity. Intriguingly, in both Torin1 and control treated cells phosphorylation of the ULK1 substrate ATG13 at serine (S) 318 was substantially increased upon SMCR8 knockdown (*Figure 7A*), while lack of C9ORF72 or WDR41 caused the opposite effect (*Figure 7B and C*). Monitoring S29 phosphorylation of ATG14, which represents another substrate of ULK1, independently confirmed the inhibitory function of SMCR8 on ULK1 kinase activity (*Figure 7D* and *Figure 7—figure supplement 1A*), whereas absence of C9ORF72 left S29 phosphorylation unchanged (*Figure 7E*). Since ULK1 kinase activity can be regulated via several upstream kinases (*Alers et al., 2012*), we examined whether SMCR8 mediated repression of ULK1 kinase activity is dependent on mTORC1 or AMPK. As expected, Torin1 treatment completely blocked mTORC1-dependent ULK1 S757 phosphorylation and led to concurrently increased phosphorylation of ATG13 (*Figure 7A,C and F* (compare sicon DMSO to sicon Torin1)). Similarly, glucose starvation increased AMPK-dependent ULK1 S317 phosphorylation (*Figure 7—figure supplement 1B*). Upon SMCR8 depletion, mTORC1-dependent phosphorylation of ULK1 at S757 was completely abolished after treatment with Torin1 but enhanced in control cells (*Figure 7F*). Unexpectedly, absence of SMCR8 increased ULK1 protein levels more than 3-fold compared to control cells. Densitometric analysis of the ratio between phosphorylated and total ULK1 protein levels revealed that S757 phosphorylation is marginally decreased upon SMCR8 knockdown. A similar phenotype was also observed for the mTORC1 substrate S6K but not for 4EBP1. In contrast, SMCR8 depletion did not change AMPK-dependent ULK1 S317 phosphorylation (*Figure 7G*). Since ULK1 protein levels were elevated in SMCR8 depleted cells, we tested the effect of ULK1 overexpression on ATG13 phosphorylation. Increasing the amounts of ULK1 by simple overexpression slightly induced ATG13 S318 phosphorylation, while SMCR8 knockdown caused a more than 3-fold increase (*Figure 7H*), suggesting that SMCR8 imposes an inhibitory effect on the kinase activity of ULK1 in addition to controlling ULK1 protein abundance. Together with the observation that phosphorylation of ATG13 is higher in untreated cells depleted of SMCR8 than in Torin1 stimulated sicon transfected cells (*Figure 7A*), these results indicate that SMCR8-mediated regulation of ULK1 kinase activity comprises mTORC1-dependent and -independent traits.

## SMCR8 regulates ULK1 gene expression

Given that SMCR8 depletion increased ULK1 protein amounts (*Figure 7F*), we also tested other ULK1 complex components in this regard. However, in contrast to ULK1, FIP200 and ATG13 protein levels remained unchanged upon SMCR8 knockdown (*Figure 8A*). Importantly, re-expression of full-length SMCR8 was able to rescue SMCR8 depleted cells from elevated ULK1 protein levels (*Figure 8B*). Next, we investigated whether SMCR8 regulates ULK1 protein abundance in concert with its binding partners C9ORF72 and WDR41. However, ULK1 protein levels remained unchanged in cells lacking C9ORF72 or WDR41 (*Figure 8C*). Thus, regulation of ULK1 protein abundance by SMCR8 seems independent of its function within the SMCR8-C9ORF72-WDR41 GEF complex. At last, we assessed whether the SMCR8 dependent increase in ULK1 protein levels is due to altered ULK1 gene expression. RT-qPCR analysis revealed that knockdown of SMCR8 caused elevated ULK1 mRNA levels, while FIP200 mRNA levels remained unchanged, consistent with constant FIP200 protein abundance upon SMCR8 depletion (*Figure 8A,D and E*). Collectively, our data supports a dual role of SMCR8 in regulating ULK1 at the level of gene expression and kinase activity.

## Restored regulation of ULK1 protein levels in SMCR8 knockout cells

We initially confirmed the observed increase in ULK1 protein levels upon SMCR8 knockdown in HAP1 SMCR8 knockout cells (*Figure 8—figure supplement 1A*). However, later passages of these cells were devoid of any tested phenotype (*Figure 8—figure supplement 1B*). Subsequently, we generated a 293T SMCR8 knockout cell line using CRISPR-Cas9. It is noteworthy that due to clonal selection we could only start ULK1 protein expression analysis after about 6 weeks. Despite SMCR8 deletion, ULK1 protein levels again remained unchanged (*Figure 8—figure supplement 1C*). For a time-resolved segmentation, the ULK1 protein abundance was monitored in absence of SMCR8 with a long-term siRNA knockdown experiment: First, 293 T cells were transfected with non-targeting or SMCR8 siRNA. After 2–3 days half of the cells were re-transfected with siRNA while the other half

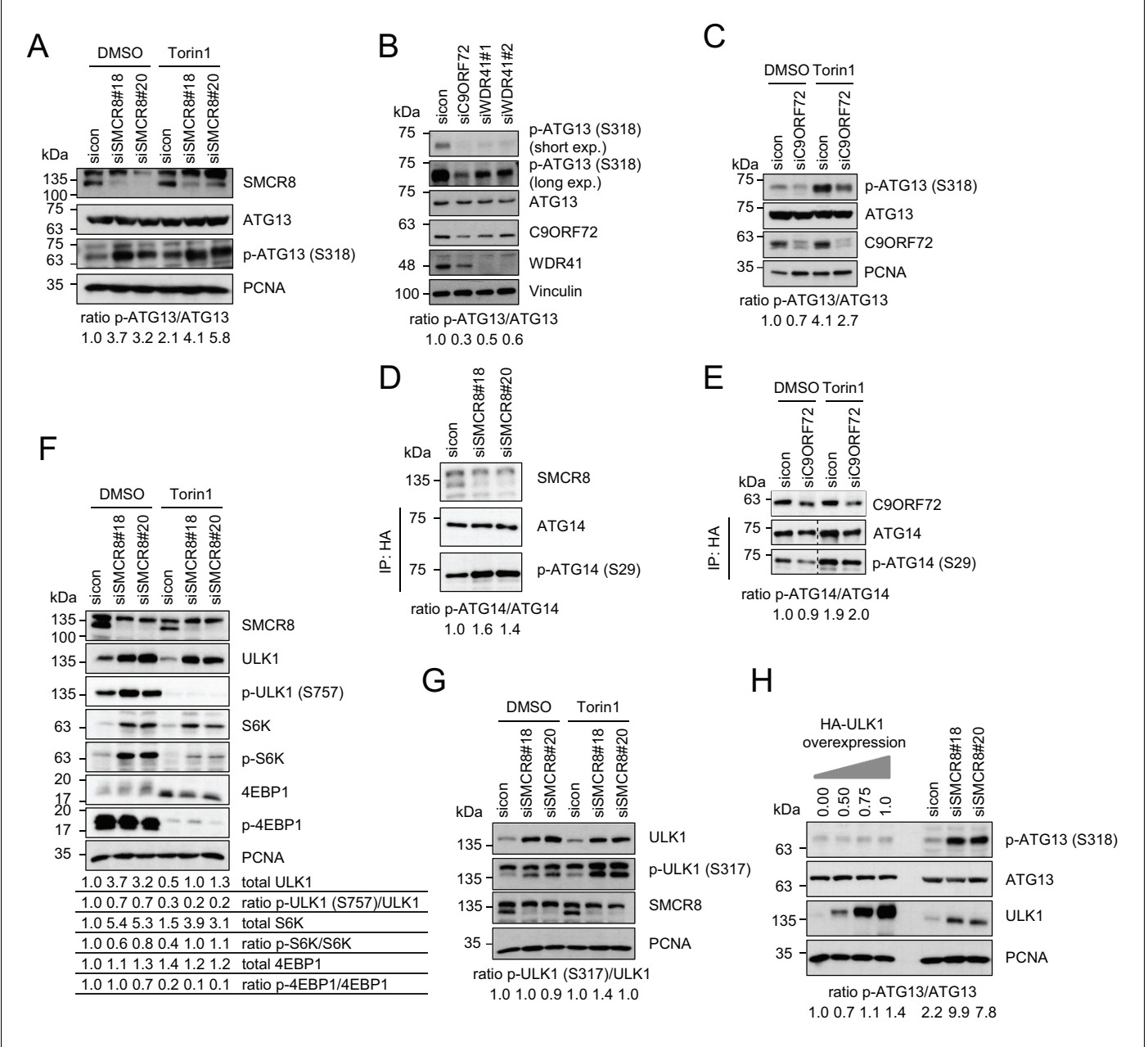

**Figure 7.** SMCR8 regulates ULK1 kinase activity. (**A**) Lysates from 293 T cells transfected with non-targeting control (sicon) or SMCR8 siRNA and grown in absence (DMSO) or presence of 250 nM Torin1 for 2 hr were subjected to SDS-PAGE and immunoblotting with indicated antibodies. PCNA served as loading control. Immunoblots were quantified using ImageJ and the ratio of p-ATG13/ATG13 was calculated. (**B**) Lysates from 293 T cells transfected with non-targeting control (sicon), C9ORF72 or WDR41 siRNAs were subjected to SDS-PAGE and immunoblotting with indicated antibodies. Vinculin served as loading control. exp. = exposure. Immunoblots were quantified using ImageJ and the ratio of p-ATG13/ATG13 was calculated. (**C**) Lysates from 293 T cells transfected with non-targeting control (sicon) or C9ORF72 siRNA and grown in absence (DMSO) or presence of 250 nM Torin1 for 2 hr were subjected to SDS-PAGE and immunoblotting with indicated antibodies. PCNA served as loading control. Immunoblots were quantified using ImageJ and the ratio of p-ATG13/ATG13 was calculated. (**D**) Lysates from 293 T cells transfected with non-targeting control (sicon) or SMCR8 siRNA and HA-tagged ATG14 were subjected to HA-IP followed by SDS-PAGE and immunoblotting with indicated antibodies. Immunoblots were quantified using ImageJ and the ratio of p-ATG14/ATG14 was calculated. (**E**) Lysates from 293 T cells transfected with non-targeting control (sicon) or C9ORF72 siRNA and HA-tagged ATG14, grown in absence (DMSO) or presence of 250 nM Torin1 for 2 hr were analyzed as in (**D**). (**F**) Cells in (**A**) were analyzed as in (**A**). Immunoblots were quantified using ImageJ. Total amounts of ULK1, S6K and 4EBP1 as well as the ratio of p-ULK1(S757)/ULK1, p-S6K/S6K and p-4EBP1/4EBP1 was calculated. (**G**) Cells in (**A**) were analyzed as in (**A**). Immunoblots were quantified using ImageJ. The ratio of p-ULK1(S317)/ULK1 was calculated. (**H**) 293 T cells transfected with non-targeting control (sicon) or SMCR8 siRNA or with increasing amounts of HA-ULK1 were lysed and analyzed as in (**A**). Immunoblots were quantified using ImageJ and the ratio of p-ATG13/ATG13 was calculated.

*Figure 7 continued on next page*

*Figure 7 continued*

The following figure supplement is available for figure 7:

**Figure supplement 1.** Evaluation of the phospho-antibody specificity.

was harvested for SDS-PAGE and immunoblotting. The same procedure was applied for several weeks. During this time course, we observed a rescue of elevated ULK1 protein levels after 4 weeks of siRNA knockdown, while SMCR8 was still depleted (*Figure 8—figure supplement 1D*). These

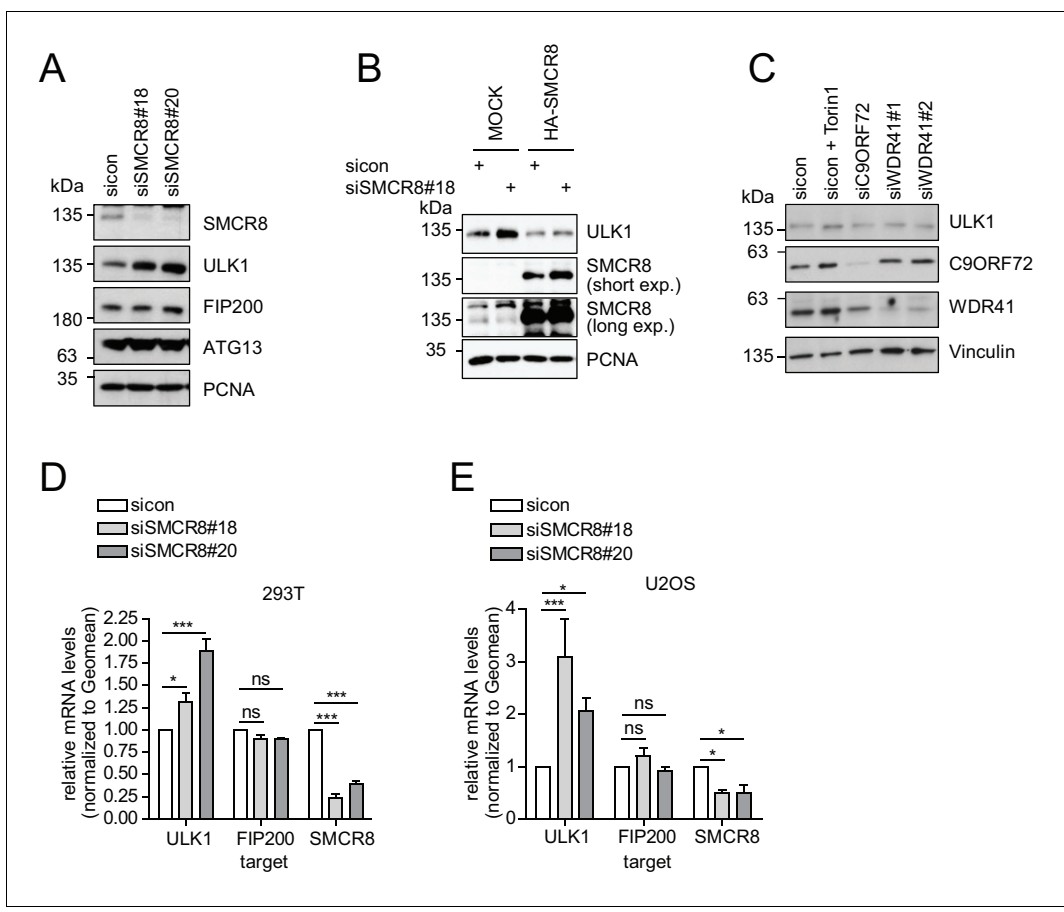

**Figure 8.** SMCR8 regulates ULK1 gene expression. (**A**) Lysates from 293 T cells transfected with non-targeting control (sicon) or SMCR8 siRNA were subjected to SDS-PAGE and immunoblotting with indicated antibodies. PCNA served as loading control. (**B**) Lysates from 293 T cells transfected with non-targeting control (sicon) or SMCR8 siRNA as well as with HA-tagged SMCR8 were subjected to SDS-PAGE and immunoblotting with indicated antibodies. PCNA served as loading control. exp. = exposure. (**C**) Lysates from 293 T cells transfected with indicated siRNAs and grown in absence (DMSO) or presence of 250 nM Torin1 for 2 hr were lysed and analyzed as in (**A**). Vinculin served as loading control. (**D,E**) 293T (**D**) or U2OS (**E**) cells were transfected with indicated siRNAs for 72 hr prior to RNA isolation, preparation of cDNA and RT-qPCR with ULK1, FIP200 and SMCR8 specific primers. Error bars represent SEM. Significance was determined using two-way ANOVA compared with sicon. All experiments were performed n = 3.

The following figure supplement is available for figure 8:

**Figure supplement 1.** Restored regulation of ULK1 protein levels in SMCR8 knockout cells.

data suggest the potential occurrence of a compensatory mechanism that restores ULK1 protein abundance in case of permanent absence of SMCR8.

## SMCR8 associates with chromatin at the gene locus of ULK1

As SMCR8 negatively controlled ULK1 mRNA levels, we examined the subcellular distribution of SMCR8 by IF using a panel of cells stably expressing HA-tagged full-length SMCR8 or fragments thereof. N- and C-terminal tagged full-length SMCR8 as well as the N-terminal fragment 1–700, which contained the binding regions for ULK1, ATG13, FIP200 and C9ORF72 (*Figure 5C*), were mainly distributed in the cytoplasm, while a minor amount of all exogenous SMCR8 variants was also located to the nucleus (*Figure 9A*, magnification). In contrast, the C-terminal SMCR8 fragment spanning aa 701–937 was mainly detected in the nucleus (*Figure 9A*). Subcellular fractionation was performed to independently confirm nuclear localization of SMCR8 and to probe its association with chromatin. Proper separation of subcellular fractions was confirmed by immunoblotting for a panel of appropriate marker proteins. FIP200 and the lysosomal protein LAMP2 localized to the cytoplasm and membrane fraction, respectively, while LaminA/C, a membrane component of the nucleus, was equally distributed between the nucleoplasm and the chromatin fraction and HistoneH3 was exclusively found in the latter (*Figure 9B*). While endogenous SMCR8 was predominantly detected in the cytoplasm and in the membrane fraction using a specific anti-SMCR8 antibody (*Figure 9B*, *Figure 9—figure supplement 1A*), subcellular fractionation followed by HA-IP of endogenously HA-tagged SMCR8 (*Figure 9—figure supplement 1B*) additionally revealed that small amounts of SMCR8 distributed to the nucleoplasm and the chromatin fraction (*Figure 9C and D*). Exogenously expressed full length SMCR8 confirmed these results (*Figure 9B*). Conversely, the N-terminal SMCR8 fragment 1–700 was almost exclusively found in the cytoplasm and membrane fraction and could not be detected on chromatin. Finally, SMCR8 fragment 701–937 was equally distributed across all fractions including chromatin (*Figure 9B*).

Given that these data provide strong evidence that SMCR8 associates with chromatin in a manner dependent on its C-terminus, we performed chromatin immunoprecipitation (ChIP) experiments to identify specific gene locus regions targeted by SMCR8. Briefly, control cells and cells expressing full-length SMCR8 or N- or C-terminal fragments thereof (*Figure 9E*) were cross-linked prior to chromatin fragmentation and anti-HA-IP. Thereafter, DNA was isolated from anti-HA immunoprecipitated chromatin and analyzed by qPCR using primers that annealed to the ULK1 or FIP200 gene locus, respectively. Intriguingly, exogenous full-length SMCR8 was significantly enriched at the ULK1 gene locus, but not at the one of FIP200 (*Figure 9F*). While the N-terminal SMCR8 fragment 1–700 did not show significant enrichment, the C-terminal SMCR8 fragment 701–937 was sufficient for the engagement of SMCR8 at the ULK1 locus and in fact was even more effective in associating with the ULK1 gene locus than exogenous full-length SMCR8 (*Figure 9F*). To confirm these findings, we performed ChIP experiments with an anti-SMCR8 antibody and unraveled significant enrichment of endogenous SMCR8 at the ULK1 gene locus. This specific chromatin association was dramatically diminished upon SMCR8 depletion (*Figure 9G and H*). In summary, SMCR8 inhibits gene expression of ULK1 dependent on its C-terminus.

## SMCR8 regulates gene expression of several autophagy genes

The regulation of ULK1 expression by SMCR8 prompted us to employ mRNA expression microarray analysis to screen for other potential transcriptional targets in an unbiased manner. Indeed, upon SMCR8 depletion the mRNA of 1059 genes were upregulated more than 1.3 fold, while 424 mRNAs showed reduced expression by more than 0.7 fold (*Figure 10A*, *Figure 10—figure supplement 1A*). Functional annotation analysis of these regulated candidate genes revealed enrichment of components of ER stress response, translation, cell cycle and DNA damage response among several other gene ontology (GO) categories (*Figure 10—figure supplement 1B and C*). Since autophagy proteins were not specifically enriched in our GO analysis, we manually curated the microarray data for mRNAs encoding proteins involved in autophagy, mTORC1 regulation and/or the lysosomal pathway (*Figure 10B*). In this data set, we detected ULK1 and S6K (RPS6KB1) among the mRNAs whose expression increased upon SMCR8 depletion, thereby confirming our initial immunoblot findings (*Figure 7F*). Using RT-qPCR, we validated several mRNA expression changes in this subset of the microarray. For example, depletion of SMCR8 led to significantly reduced mRNA levels of ATF4

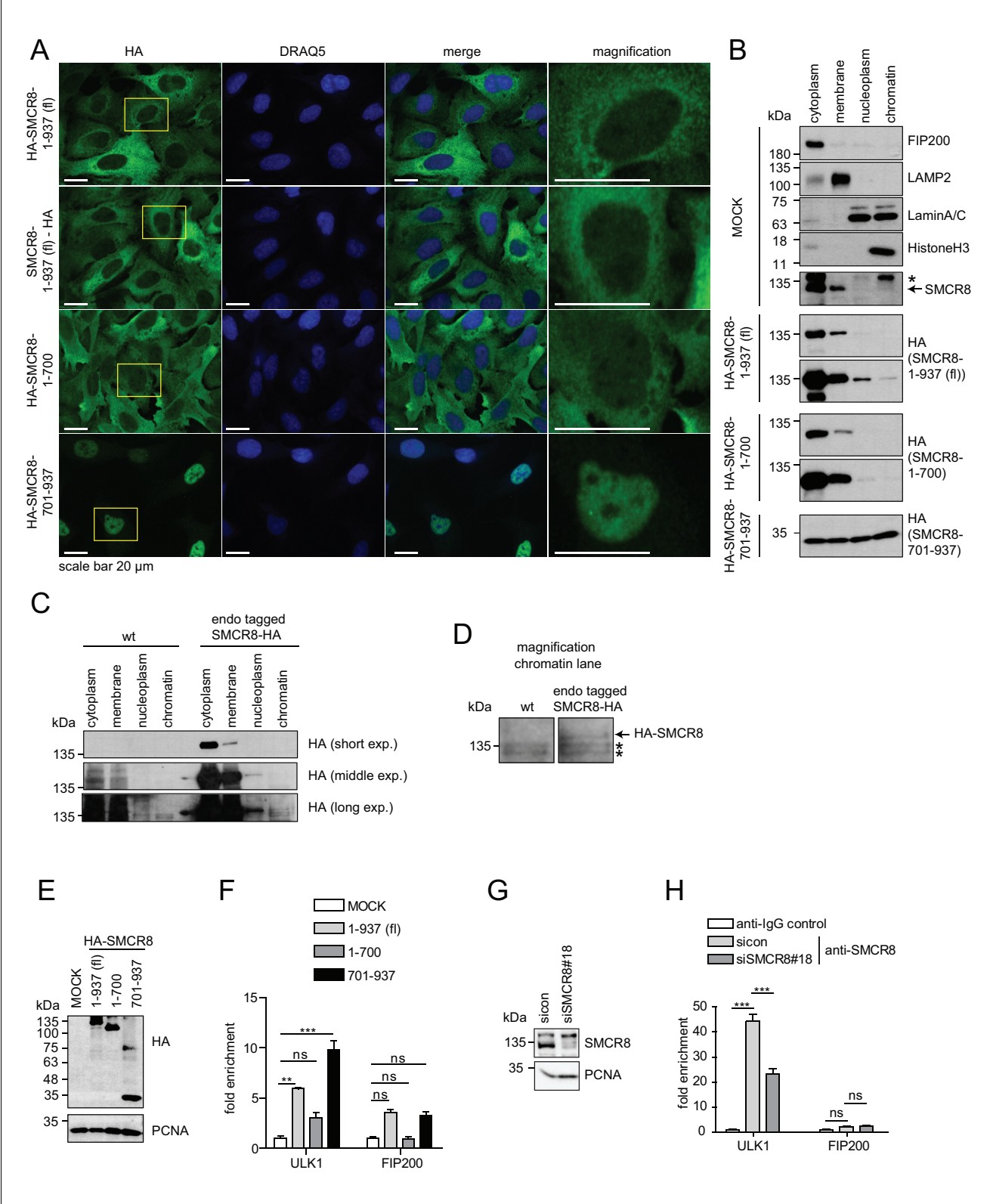

**Figure 9.** The C-terminal part of SMCR8 mediates nuclear localization and associates with the ULK1 gene locus. (**A**) U2OS cells stably expressing N- or C-terminal HA-tagged full-length (fl) SMCR8 or indicated fragments thereof were fixed and immunolabeled with anti-HA antibody. Scale bars, 20 μm. (**B**) 293 T cells were transiently transfected with HA-tagged full-length (fl) SMCR8 or indicated fragments thereof followed by subcellular fractionation, SDS-PAGE and immunoblot analysis with indicated antibodies. * or arrow indicate non-specific or specific bands, respectively. (**C**) 293 T wildtype (wt)

*Figure 9 continued on next page*

*Figure 9 continued*

cells or those with endogenously HA-tagged SMCR8 were subjected to subcellular fractionation followed by HA-IP, SDS-PAGE and immunoblot analysis with indicated antibodies. exp. = exposure. (**D**) Magnification of the chromatin lane in (**C**) for better visualization. * indicates non-specific bands. (**E,F**) Cells transfected with SMCR8 variants as in (**B**) were lysed and analyzed by SDS-PAGE and immunoblotting (**E**) or subjected to chromatin immunoprecipitation (ChIP) with an anti-HA-antibody and qPCR with primers specific for ULK1 and FIP200 (**F**). Percentages of input were calculated and normalized to MOCK. Error bars represent SEM. Significance was determined using two-way ANOVA compared with MOCK. All experiments were performed n = 3. (**G,H**) 293 T cells transfected with non-targeting (sicon) or SMCR8 siRNA for 72 hr were lysed and analyzed as in (**E**) (**G**) or subjected to ChIP with anti-SMCR8-antibody and qPCR with primers specific for ULK1 and FIP200 (**H**). Percentages of input were calculated and normalized to IgG control. Error bars represent SEM. Significance was determined using two-way ANOVA compared with sicon. All experiments were performed n = 3.

The following figure supplement is available for figure 9:

**Figure supplement 1.** Evaluation of SMCR8 antibody specificity and SMCR8 cell line.

and LAMP1, while ATG3 and ATG7 remained unchanged as in the microarray analysis (*Figure 10C*). Moreover, substantially increased mRNA levels were observed for LAMP2, S6K and WIPI2 (*Figure 10C*). The latter was also confirmed in another cell line (*Figure 10—figure supplement 1D*). Importantly, the increase or reduction in mRNA levels translated into the respective change in protein abundance in SMCR8 depleted cells (*Figure 10C*). Furthermore, subsequent ChIP experiments revealed association of HA-tagged full-length and the N-terminally truncated fragment 701–937 of SMCR8 on the WIPI2 gene locus (*Figure 10D*). As for the ULK1 gene locus these results were confirmed with anti-SMCR8 antibody at endogenous levels (*Figure 10E*). Hence, we established SMCR8 as transcriptional regulator for several autophagy genes.

Together, our findings demonstrate that SMCR8 functions as multifaceted autophagy regulator (*Figure 11*). In addition to its autophagosome maturation-promoting role as part of a GEF complex together with C9ORF72 and WDR41 (*Sellier et al., 2016*), SMCR8 impairs autophagy initiation by interacting with the ULK1 complex and inhibiting its kinase activity on one hand and associates with chromatin at the ULK1 and WIPI2 gene locus and suppresses ULK1 and WIPI2 gene expression on the other hand.

## Discussion

Using a focused image-based siRNA screen monitoring in parallel early and late autophagosomes at endogenous levels, we identified 34 out of 186 members of the Rab GTPase, GAP and GEF families that function in autophagy. Based on ultrastructural and interaction network analysis we decided to further investigate SMCR8. In summary, we confirmed and extended recent findings that SMCR8 regulates the autophagosomal-lysosomal pathway and associates with the ULK1 complex and C9ORF72 (*Sellier et al., 2016*; *Amick et al., 2016*; *Sullivan et al., 2016*; *Yang et al., 2016*; *Xiao et al., 2016*; *Blokhuis et al., 2016*; *Ugolino et al., 2016*).

Our SMCR8 interaction studies revealed that ULK1 complex components and C9ORF72 employ overlapping binding regions for their association with SMCR8. Particular interesting is that ATG13 and C9ORF72 show differential binding to SMCR8 in the region spanning aa 320–400. This raises the possibility that association of ATG13 (together with ULK1, FIP200 and ATG101) and C9ORF72 (together with WDR41) with SMCR8 is potentially distinctively regulated. Intriguingly, autophagy induction left the SMCR8 interaction with C9ORF72 unimpaired, while association of both with the ULK1 complex increased substantially. However, neither did ATG13 overexpression disrupt association between SMCR8 and C9ORF72, nor changed the ULK1 complex during SMCR8 overexpression or depletion. Together with our SEC and Native PAGE analysis, these data indicate the co-existence of a separate SMCR8-C9ORF72-WDR41 complex and a combined SMCR8-C9ORF72-WDR41-ULK1 complex holo-assembly, which might preferentially form after autophagy induction although we did not observe major changes in the holo-assembly composition upon Torin1 treatment.

Intriguingly, we found that depletion of SMCR8 impaired both autophagosome formation and maturation. This phenomenon has previously been described for RAB11 (*Longatti et al., 2012*; *Fader et al., 2008*), which inhibits autophagosome formation together with TBC1D14 by mediating transport and fusion events of endosomes (*Longatti et al., 2012*; *Fader et al., 2008*). Another

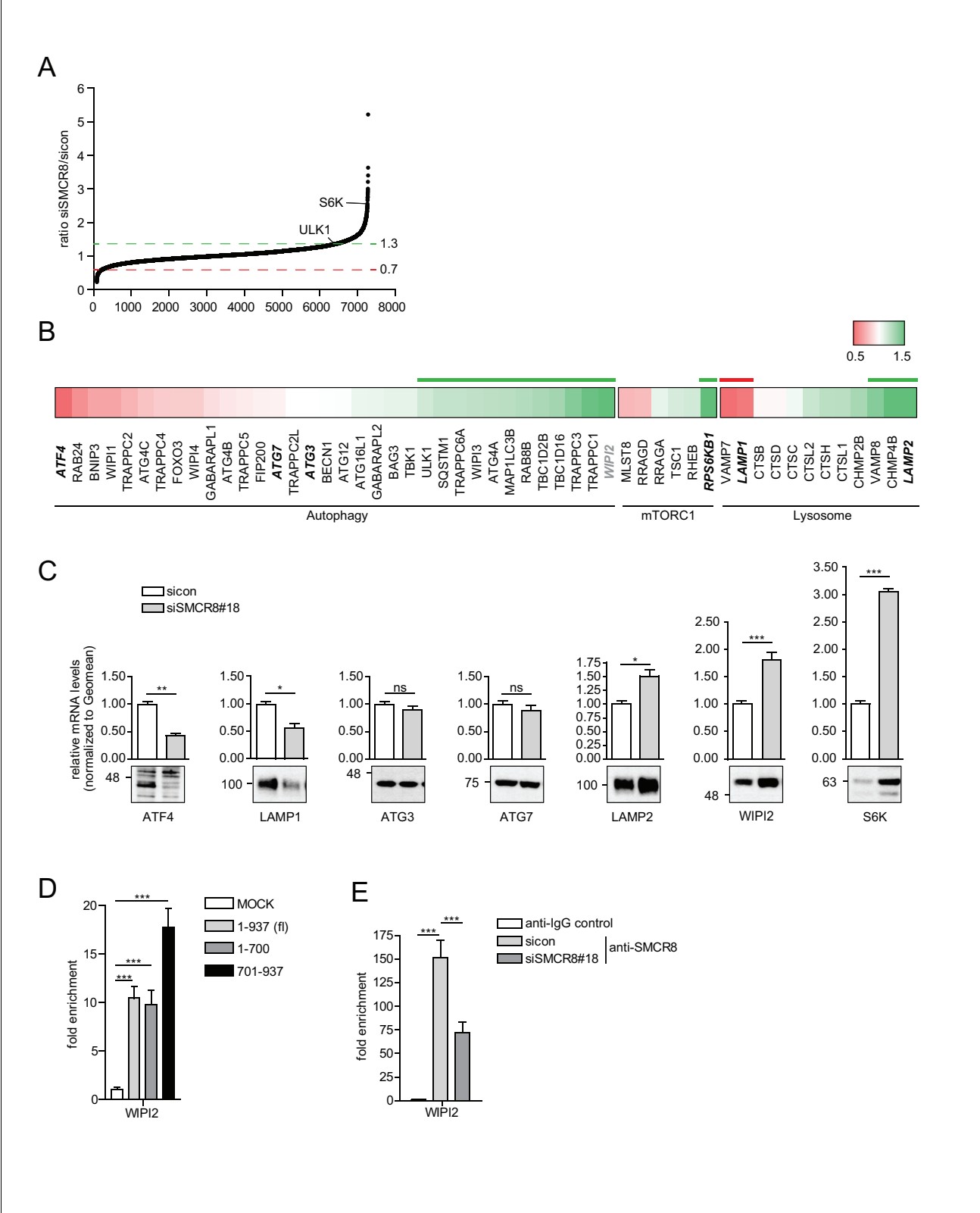

**Figure 10.** SMCR8 regulates gene expression of autophagosomal proteins. (**A**) 293 T cells were transfected with non-targeting control (sicon) or SMCR8 siRNA prior to RNA isolation and microarray analysis. Representation of normalized ratios of siSMCR8/sicon of three independent experiments. See *Figure 10—source data 1* for complete microarray analysis. (**B**) Selected autophagosomal and lysosomal genes from data in (**A**) are shown as heatmap representation. Genes upregulated more than 1.3 fold or downregulated more than 0.7 fold are marked with a green or red bar, respectively. Genes

*Figure 10 continued on next page*

*Figure 10 continued*

selected for validation are marked in bold and italic. WIPI2 is marked in grey, due to our stringent quality control. (C) 293 T cells were transfected with non-targeting control (sicon) or SMCR8 siRNA for 72 hr prior to RNA isolation, preparation of cDNA and RT-qPCR with indicated specific primers or subjected to SDS-PAGE and immunoblotting with indicated antibodies. Error bars represent SEM. Significance was determined using unpaired t-test. All experiments were performed n = 3. (D) 293 T cells transiently transfected with HA-tagged full-length (fl) SMCR8 or indicated fragments thereof were lysed and subjected to chromatin immunoprecipitation (ChIP) with anti-HA-antibody and qPCR with primers specific for WIPI2. Percentages of input were calculated and normalized to MOCK. Error bars represent SEM. Significance was determined using one-way ANOVA compared with MOCK. All experiments were performed n = 3. (E) 293 T cells transfected with non-targeting (sicon) or SMCR8 siRNA for 72 hr were lysed and subjected to ChIP with an anti-SMCR8-antibody and qPCR with primers specific for WIPI2. Percentages of input were calculated and normalized to IgG control. Error bars represent SEM. Significance was determined using one-way ANOVA compared with sicon. All experiments were performed n = 3.

The following source data and figure supplement are available for figure 10:

**Source data 1.** mRNA expression microarray analysis of control and SMCR8 depleted cells.

**Figure supplement 1.** SMCR8 regulates gene expression of autophagosomal proteins.

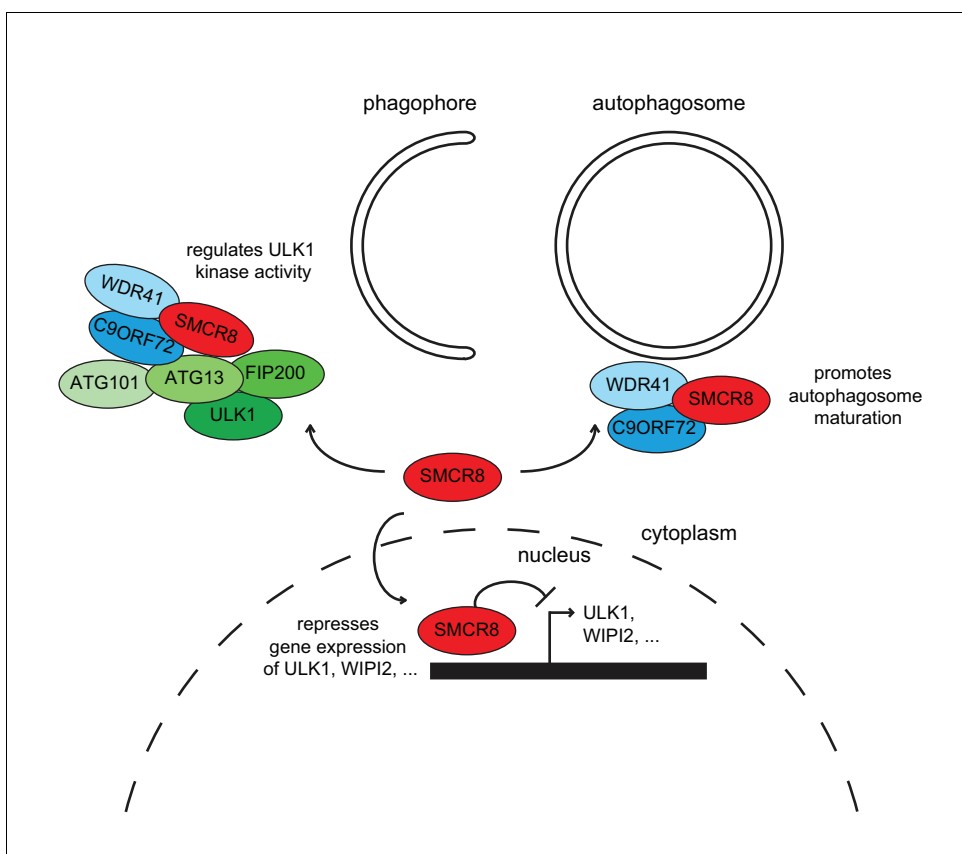

**Figure 11.** Working model for the multifaceted function of SMCR8 during autophagy. On one hand, SMCR8 promotes autophagosome maturation as part of a trimeric RAB39B GEF complex together with C9ORF72 and WDR41 as previously shown *Sellier et al. (2016)*. On the other hand, this SMCR8 complex regulates autophagosome formation by binding ULK1 complex components and modulating the kinase activity of ULK1. Furthermore, SMCR8 associates with the ULK1 and WIPI2 gene locus and represses ULK1 and WIPI2 gene expression and additionally regulates transcription of several other autophagy-related genes.

example is RAB33B, which first accelerates autophagosome formation by recruitment of the ATG8 lipidation machinery (*Fukuda and Itoh, 2008*; *Itoh et al., 2008*) and thereafter autophagosome fusion with lysosomes (*Itoh et al., 2011*).

Regulation of phagophore initiation is highly dependent on phosphorylation events (*Stork et al., 2012*). For example, increased phosphorylation of ULK1 kinase substrates promotes autophagy initiation and elongation. In our study, we uncovered that SMCR8 depletion enhances ULK1-dependent ATG13 and ATG14 phosphorylation, while C9ORF72 knockdown surprisingly had the opposing effect on ATG13 and no effect on ATG14. Given that both are found associated with the ULK1 complex it is conceivable that ULK1 kinase activity regulation is due to direct binding of SMCR8 and/or C9ORF72 to ULK1 and/or its complex partners. However, SMCR8 overexpression or depletion does not disrupt association of ULK1 and ATG13 or induce changes in the ULK1 complex fractionation pattern. Since C9ORF72 recruits the ULK1 complex to the nascent phagophore (*Webster et al., 2016*), the localization of the ULK1 complex could presumably also be linked to its activation. Another potential regulatory mechanism to control ULK1 kinase activity is ULK1 phosphorylation via upstream kinases such as mTORC1 and AMPK (*Egan et al., 2011*). We observed a reduction in phosphorylation of the mTORC1 substrates ULK1 and S6K, which are simultaneously upregulated at the transcriptional level in SMCR8 knockdown cells. Notably, phosphorylation status of another mTORC1 substrate, 4EBP1, remained unchanged. Furthermore, AMPK kinase activity was unimpaired in respect to ULK1 phosphorylation. Together, these findings indicate that SMCR8 controls ULK1 activity via mTORC1 dependent and independent pathways. Further detailed studies in vitro are required to mechanistically dissect how SMCR8 and C9ORF72 modulate ULK1 kinase activity in a substrate-specific manner.

SMCR8 itself was reported to be phosphorylated by several kinases including AMPK, mTORC1, ULK1 and TBK1 (*Hsu et al., 2011*; *Sellier et al., 2016*; *Hoffman et al., 2015*). The latter was recently shown to regulate the GEF activity of SMCR8 towards RAB39B (*Sellier et al., 2016*). In addition, phosphorylation of SMCR8 could potentially regulate binding of SMCR8 to the ULK1 complex. Alternatively, SMCR8 phosphorylation might play a role in controlling the distribution of SMCR8 between the cytoplasm and nucleus in a similar manner as shown for the transcription factor TFEB (*Settembre et al., 2012*).

Unexpectedly, we unraveled that the recently observed increased ULK1 protein abundance upon SMCR8 depletion (*Yang et al., 2016*) was due to increased ULK1 mRNA levels. This phenotype was independent of C9ORF72 and required the C-terminal part of SMCR8 spanning aa 701–937 as this fragment was almost exclusively localized to the nucleus and enriched at the ULK1 gene locus to a substantial higher level than full-length SMCR8. Since SMCR8 lacks a clear nuclear localization sequence or export signal, future functional analyses will need to address whether self-inhibitory and/or phosphorylation-dependent mechanisms regulate nuclear translocation of SMCR8. Similarly, SMCR8 directly associates with chromatin but does not contain a bioinformatically detectable DNA binding domain. Thus, it is likely that SMCR8 represses ULK1 gene expression through interaction with another chromatin-associated protein. Recently, STRaNDs were defined as novel group of non-DNA binding, cytoplasmic proteins, which shuttle into the nucleus and regulate gene expression through interaction with transcription factors (*Lu et al., 2016*). In this regard, potential STRaND cooperation partners of SMCR8 are ATF4, p53, FOXO3 and ZKSCAN3, which all regulate mRNA expression of ULK1 and several other autophagosomal and lysosomal proteins (*Settembre et al., 2011*; *Chauhan et al., 2013*; *Chua et al., 2014*; *Pietrocola et al., 2013*). Along this line, our global mRNA expression analysis revealed that SMCR8 controls gene expression of multiple autophagosomal and lysosomal proteins, among them WIPI2.

Finally, hexanucleotide expansion mutation in the 5' UTR of *C9ORF72* causes amyotrophic lateral sclerosis (ALS) and frontotemporal dementia (FTD) (*Salameh et al., 2015*; *Weder et al., 2007*; *Cruts et al., 1993*). Interestingly, C9ORF72 was one of the validated candidates in our screen and recently implemented in RAB1A dependent recruitment of the ULK1 complex to the phagophore (*Webster et al., 2016*). Consistently, RAB1A depletion decreased number of WIPI2 positive spots in our primary screen but did not fulfill our stringent standard deviation criterion to be included in the deconvolution screen. Concurrent with the SMCR8-C9ORF72-WDR41 complex possessing GEF activity towards RAB39B and thereby regulating autophagosome maturation (*Sellier et al., 2016*), we identified RAB39B as candidate in our primary screen. However, RAB39B was excluded from further analysis, since it was outranked by other candidates. While ULK1 kinase activity is regulated by both

SMCR8 and C9ORF72, we found that ULK1 gene repression is seemingly independent of the SMCR8-C9ORF72-WDR41 GEF complex since ULK1 protein levels remained unchanged in cells lacking C9ORF72 or WDR41. Furthermore, the C-terminal fragment of SMCR8, which does not bind C9ORF72, was sufficient to associate with chromatin at the ULK1 and WIPI2 gene locus. Intriguingly, SMCR8 regulated gene expression of several autophagosomal but also lysosomal proteins, such as LAMP1 and LAMP2. Since SMCR8 and C9ORF72 protein levels are interdependent (*Amick et al., 2016*) and lysosomal dysfunction was detected in SMCR8 ko cells as well as in C9ORF72 ko mice (*Amick et al., 2016*; *Sullivan et al., 2016*), future studies are required to reveal whether SMCR8 plays a role in ALS-FTD alongside with C9ORF72.

# Materials and methods

## Antibodies

Following antibodies were used: Anti-4EBP1 (Cell Signaling, Danvers, MA, #9644, RRID: AB_2097841); anti-phospho-4EBP1 (S65 Cell Signaling #9451, RRID:AB_330947); anti-ATF4 (Cell Signaling #11815, RRID:AB_2616025); anti-ATG2B (Sigma, St. Louis, MO, A96430); anti-ATG3 (Cell Signaling #3415, RRID:AB_2059244); anti-ATG7 (Cell Signaling #8558, RRID:AB_10831194); anti-ATG12 (Cell Signaling #2010, RRID:AB_2059086); anti-ATG13 (MBL, Woburn, MA, M183-3, RRID:AB_10796107); anti-phospho-ATG13 (Ser318 Rockland, Limerick, PA, 600–401 C49, RRID:AB_11179920); anti-ATG14 (Cell Signaling #5504, RRID:AB_10695397); anti-phospho-ATG14 (S29 Cell Signaling #13155); anti-C9ORF72 (Santa Cruz, Dallas, TX, sc138763, RRID:AB_10709750); anti-FIP200 (Proteintech, Rosemont, IL, 17250–1-AP, RRID:AB_10666428); anti-flag (Cell Signaling #2368, RRID:AB_2217020); anti-GABARAP (Abcam, Cambridge, MA, ab109364, RRID:AB_10861928); anti-HA (Covance, Princeton, NJ, MMS-101P, RRID:AB_2314672; Roche, Basel, Switzerland, 11867423001, RRID:AB_390918; Abcam ab9110, RRID:AB_307019); anti-HistoneH3 (Abcam ab1791, RRID:AB_302613); anti-myc (Santa Cruz sc788, RRID:AB_631277); anti-LAMP1 (DSHB, Iowa City, IA, H4A3, RRID:AB_2296838); anti-LAMP2 (Abcam ab25631, RRID:AB_470709); anti-LaminA/C (Epitomics, Burlingame, CA, 2966–1, RRID:AB_2136262); anti-LC3B (Cell Signaling #2775, RRID:AB_915950; MBL PM036, RRID:AB_2274121); anti-RAB7A (Cell Signaling #2094, RRID:AB_2300652); anti-PCNA (Santa Cruz sc-7907, RRID:AB_2160375); anti-PIK3C3 (Cell Signaling #3358, RRID:AB_10828387); anti-S6K (Cell Signaling #9202, RRID:AB_331676); anti-phospho-S6K (T389 Cell Signaling #9234, RRID:AB_2269803); anti-SMCR8 (Abcam ab202283); anti-STX17 (Sigma HPA001204, RRID:AB_1080118); anti-ULK1 (Cell Signaling 8054, RRID:AB_11178668); anti-phospho-ULK1 (S317 Cell Signaling #12753); anti-phospho-ULK1 (S757 Cell Signaling #6888, RRID:AB_10829226); anti-Vinculin (Sigma V4505, RRID:AB_477617); anti-VMP1 (Cell Signaling #12978); anti-WIPI2 (Abcam ab105459, RRID:AB_10860881), anti-WDR41 (Abcam ab108096, RRID:AB_10864252).

## Plasmids

PCR products generated from ORFs (obtained from the human ORFeome collection) were cloned into Gateway pDONR223 entry vector. After sequence verification cDNAs were subcloned into Gateway destination vectors for mammalian expression. The pHAGE-N-Flag-HA, pHAGE-N-GFP and MSCV-i(N-Flag-HA)-IRES-PURO vectors were used for transient transfection of 293 T or 293T-REx cells. Moreover, stable cells were generated by retroviral transduction of MSCV-i(N-Flag-HA)-IRES-PURO or lentiviral transduction of pHAGE-N-Flag-HA or pHAGE-C-Flag-HA followed by selection with antibiotics.

## Cell culture

HEK-293 T (RRID:CVCL_0063), HEK-293T-REx (RRID:CVCL_D585) and U2OS (RRID:CVCL_0042) cells were cultured in Dulbecco's modified Eagle's medium (DMEM, Life Technologies/ Thermo Fisher Scientific, Waltham, MA), while HAP1 cells were cultured in Iscove's modified Dulbecco's medium (IMDM, Life Technologies), all supplemented with 10% fetal bovine serum (FBS), 2 mM glutamine and antibiotics (Puromycin (2 μg/ml, Life Technologies), Blasticidin (4–15 μg/ml, Invivogen, San Diego, CA) or Geneticin (600 μg/ml, Life Technologies)) as necessary and maintained at 37°C and 5% $CO_2$. Torin1 (Tocris, Bristol, UK; 250 nM) or BafilomycinA1 (Biomol, Hamburg, Germany; 100 nM) were applied to cells for 1–2 hr to modulate autophagy. In addition, autophagy was induced via

glucose starvation with DMEM (-) Glucose (Life Technologies) or complete starvation with EBSS (Life Technologies) typically for 2 hr or indicated time points. Expression of HA-tagged proteins was induced for 24 hr to 48 hr by addition of 4 µg/ml doxycycline (Sigma) in stable cells or by transient transfection (see below). HEK-293T, HEK-293T-REx and U2OS cells were purchased from ATCC, Manassas, VA. Human HAP1 SMCR8 knockout cells were purchased from Horizon Discovery, Waterbeach, UK, (HZGHC003606c011). All cell lines were regularly tested negative for mycoplasma.

## Transfection-based experiments
Cells were reverse transfected with siRNAs (Dharmacon, Lafayette, CO, or Eurofins MWG Operon, Luxembourg) using Lipofectamine RNAiMax (Life Technologies) according to manufacturer's instructions and typically harvested 72 hr after transfection. siRNA sequences are listed in *Supplementary file 2*. Plasmids were transfected using Lipofectamine 2000 (Life Technologies), GeneJuice (Merck Millipore, Darmstadt, Germany) or PEI (Polyethylenimine, Polysciences Europe GmbH, Hirschberg an der Bergstrasse, Germany) according to standard protocols.

## Generation of endogenously HA-tagged SMCR8 cells via CRISPR-Cas9
C-terminal tagging of the endogenous SMCR8 gene locus via CRISPR-Cas9 (*Stewart-Ornstein and Lahav, 2016*) started with cloning of SMCR8 guide RNA sequences (gRNA-for: CACCGTGACCAA-GACCTGTGACTCA, gRNA-rev: AAACTGAGTCACAGGTCTTGGTCAC) into a Cas9 expressing plasmid (px330). This plasmid was transfected into 293 T cells together with a homology donor (100 base pairs of the SMCR8 C-terminus, mRUBY3, HA-tag, blasticidin resistance) amplified by PCR. Cells were selected using the introduced antibiotic resistance. Proper locus insertion in single clones was confirmed on genomic DNA (PureLink Genomic DNA Extraction Kit, Invitrogen/ Thermo Fisher Scientific) by PCR with locus specific primers, followed by sequencing as well as SDS-PAGE and immunoblot.

## Generation of SMCR8 knockout cell lines
Primers encompassing guideRNA sequences for SMCR8 (gRNA#1: CACCGCCTTACCCTATAC-GACCTGG, #2: CACCGATCCACAGACATGATACGCA, #3: CACCGTGCCCCTTCAACTTCCGATG) were ligated with T4 ligase into a CRISPR-Cas9 vector (pLenti2.0), which was already digested by the restriction enzyme BsmBI according to manufacturer's protocols. Guide RNA containing pLenti2.0 was verified by sequencing and transfected together with lentiviral packaging plasmids into 293 T cells as described above. Virus was harvested and applied to transduce 293 T cells. Subsequently, cells were selected with antibiotics and SMCR8 knockout in single clones was confirmed by immunoblot.

## siRNA screen
The multiplex image-based autophagy RNAi screen is described in more detail at Bio-protocol (*Jung and Behrends, 2017*). The target gene siRNA library (siON-TARGET, Dharmacon; pooled or individual siRNAs, as indicated) was distributed in 384 well imaging plates (CellCarrier-384 Black, Perkin Elmer, Waltham, MA) using a semi-automated pipettor (CyBi-SELMA). Thereafter, 1500 U2OS cells were reverse transfected using Lipofectamine RNAiMAX (Life Technologies) according to manufacturer's instructions. 72 hr after transfection, cells were fixed with 4% paraformaldehyde. See *Supplementary file 1* for siRNA sequences.

## Immunofluorescence
After fixation with 4% paraformaldehyde, cells were permeabilized with 0.5% Triton-X 100 in PBS (10 min), followed by blocking with 1% BSA in PBS for 1 hr. Primary and secondary antibodies as well as nuclear and cytoplasmic staining reagents (AlexaFluor-coupled antibodies (Life Technologies); DRAQ5 (Cell Signaling); HSC CellMask Deep red stain (Life Technologies)) were incubated in 0.1% BSA in PBS for 1 hr with three washes of PBS in between. For double stainings, antibodies were incubated sequentially.

## Image acquisition and analysis

Images were acquired on PerkinElmer's Opera High Content Screening System with a 60x water-immersion objective and analyzed with Acapella High Content Imaging Analysis Software (PerkinElmer). Image segmentation started by detection of the cell nuclei and the cytoplasm in the 633 nm channel (DRAQ5 and HSC CellMask). Cytosolic spots were determined in the 488 nm channel by using specific characteristics such as spot intensity, area and contrast. Resulting output parameters included number of spots and ISS (integrated spot signal) per cell as well as number of cells per well. Raw data of quadruplicates was averaged and subsequently normalized to non-targeting control siRNA (sicon) for every 384 well plate in Excel. To classify candidates in the primary screen, pooled siRNAs had to differ in both spot parameters (number and ISS) for two or three standard deviations from the normalized sicon depending on the autophagosome marker (WIPI2 and ATG12 = 3; LC3B, GABARAP and STX17 = 2). Parallel raw data normalization using the z-score and B-score method resulted in similar candidates and additional candidates were included. The top ten increasing and decreasing candidates that were specific for one or common for several autophagosome markers were selected for the deconvolution screen (in total 71). Then, four individual siRNAs per genes were used and validated candidates were determined by differing from sicon in the standard deviation criterion for three out of four siRNAs per gene. Toxic siRNAs were excluded based on obvious changes in number of cells as well as in the intensity and area of the nucleus and of the cytoplasm. Then, two out of three siRNAs were sufficient to determine a validated candidate gene. Genes with more than one cytotoxic siRNA were removed from further analysis.

## Immunoblotting

Cells were lysed in RIPA (50 mM Tris [pH 7.5]; 150 mM NaCl; 1% NP40; 0.1% SDS; 0.5% sodium desoxycholate) or MCLB (50 mM Tris [pH 7.4]; 150 mM NaCl; 0.5% NP40) buffer supplemented with complete EDTA-free protease inhibitor (Roche) and phosphatase inhibitor (PhosSTOP, Roche) tablets followed by addition of 4x laemmli buffer after removal of cell debris by centrifugation. Proteins were separated by SDS-PAGE (4–20% gels (BioRad, Hercules, CA) or self-casted 8% and 12% gels) and transferred to nitrocellulose (NitroBind 0.45 μm, Thermo Fisher Scientific) of PVDF (Merck Millipore) membranes, which were blocked with TBS-T (20 mM Tris; 150 mM NaCl; 0.1% Tween-20) containing 5% BSA (Sigma) or 5% low fat milk (Roth, Karlsruhe, Germany). Blots were incubated with primary antibodies in blocking buffer at 4°C overnight and secondary antibodies (anti-mouse-HRP (Promega, Madison, WI); anti-rabbit-HRP (Promega); anti-rabbit-LC-kappa (Abcam ab99617); anti-rat-HRP (Dianova, Hamburg, Germany)) were added for 1 hr after washing with TBS-T.

## RNA isolation, cDNA synthesis and real time quantitative PCR

Total RNA from U2OS or 293 T cells was isolated using High Pure RNA isolation kit (Roche) and then reverse transcribed into cDNA with Transcriptor First Strand cDNA Synthesis Kit (Roche). Real time quantitative PCR was performed on a Light Cycler 480 (Roche) employing LightCycler 480 SYBR Green I Master with specific target gene primers (*Supplementary file 2*). Relative target gene mRNA expression was normalized to the geometrical mean of three reference genes (ACTB, HMBS, and TBP).

## Immunoprecipitation

Frozen cell pellets were lysed for 30 min in ice-cold MCLB supplemented with protease and phosphatase inhibitors and cell debris was removed from lysates by centrifugation. The supernatant was subjected to immunoprecipitation with pre-equilibrated anti-HA-agarose (Sigma) overnight at 4°C. Afterwards, agarose beads were washed three times with MCLB buffer and bound proteins were eluted by addition of 4x laemmli buffer and boiling at 95°C for 5 min. Samples were then analyzed by SDS-PAGE and immunoblotting.

## Endogenous immunoprecipitation

293 T cells were lysed in MCLB buffer with protease and phosphatase inhibitors for 30 min on ice. Cell debris was removed by centrifugation and lysates were precleared by addition of Protein A/G Plus Agarose beads (Santa Cruz) for one hour at 4°C. Precleared lysates were incubated with indicated antibodies over night at 4°C followed by addition of agarose beads for 2 hr. After washing

with MCLB buffer for three times, proteins were eluted by addition of 4x laemmli buffer and boiling at 95°C for 5 min. Proteins were separated by SDS-PAGE and analyzed by immunoblotting.

## Lambda phosphatase treatment

Cells were lysed with MCLB buffer without phosphatase inhibitors followed by debris removal via centrifugation and immunoprecipitation with pre-equilibrated anti-HA-beads overnight. Then, beads were washed with MCLB buffer for three times and incubated with Lambda Protein Phosphatase (PPase, New England Biolabs, Ipswich, MA) for 1 hr according to manufacturer's instructions, prior to elution with 4x laemmli buffer and boiling at 95°C for 5 min. Samples were then analyzed by SDS-PAGE and immunoblotting.

## Mass spectrometry (MS)-based proteomics

HA-immunoprecipitation followed by MS analysis was performed as previously described (*Jung et al., 2015*; *Behrends et al., 2010*; *Sowa et al., 2009*; *Huttlin et al., 2010*). Briefly, 293T-REx cells expressing HA-tagged proteins were lysed with ice-cold MCLB buffer, cleared through 0.45 µm spin filters (Merck Millipore) and immunoprecipitated using anti-HA-agarose (Sigma). After intensive washing, proteins were eluted with HA peptide (250 µg/ml, Sigma) and precipitated with trichloroacetic acid (Sigma), followed by digestion with trypsin (Promega) and desalting by custom-made stage tips. Samples were analyzed in technical duplicates on a LTQ Velos (Thermo Fisher Scientific) and spectra were identified as previously described (*Huttlin et al., 2010*). For CompPASS analysis, we employed 142 unrelated bait proteins that were all previously processed in the same way (*Behrends et al., 2010*; *Sowa et al., 2009*). Weighted and normalized D-scores ($WD^N$-score) were calculated based on average peptide spectral matches (APSMs). Proteins with $WD^N \geq 1$ and APSM $\geq 2$ were considered as high-confident candidate interacting proteins (HCIPs) and visualized using Cytoscape.

## Native PAGE with subsequent in-gel trypsin digestion

Cells were lysed with MCLB and subjected to immunoprecipitation with HA-beads as described above. Proteins were eluted with HA-peptide in PBS and NativePAGE sample buffer (Thermo Fisher Scientific), prior to Native PAGE (NativePAGE Novex 3–12% Bis-Tris Protein Gels, NativePAGE Running Buffer, Thermo Fisher Scientific). Subsequently, gels were either immunoblotted or fixed prior to in-gel tryptic digestion for MS analysis. Briefly, gels were cut into single lanes and each lane into eight pieces. Next, gel pieces were washed three times with 50 mM ammonium bicarbonate (ABC) containing 50% ethanol followed by dehydration for 10 min with ethanol and reduction for 45 min at 56°C with 10 mM DTT in 20 mM ABC. For alkylation gel pieces were incubated with 55 mM iodoacetamide in 20 mM ABC for 30 min in the dark, washed two times with 5 mM ABC containing 50% ethanol, followed by dehydration with ethanol and consequent vacuum centrifugation. Subsequently, gel pieces were incubated with 12.5 ng/µl trypsin in 20 mM ABC overnight and eluted three times with increasing ACN concentrations. Samples were desalted via stage tips as described above. Mass spectra were obtained on a Q Exactive HF (Thermo Fisher Scientific) and analyzed using MaxQuant 1.5.3.30.

## Size exclusion chromatography

Whole cell lysates were generated via three freeze-thaw cycles in running buffer (50 mM TRIS [pH 7.5], 150 mM NaCl) and subsequent centrifugation, while HA-IP samples were prepared and eluted as described above. 500 µl sample was injected into a 500 µl loop of the ÄKTApurifier with a Superose 6 10/300 GL column (GE Healthcare, Chicago, IL) and eluted at a flow rate of 0.4 ml per min using running buffer. 500 µl fractions were collected in a 96 well plate and analyzed by SDS-PAGE and immunoblotting or MS analysis after TCA precipitation, trypsin digestion and desalting as described above. The column was calibrated with HMW and LMW Gel Filtration Calibration Kits (GE Healthcare).

## Subcellular fractionation

293 T cells were subjected to subcellular fractionation with a Subcellular Protein Fractionation Kit (Thermo Fisher Scientific) according to manufacturer's instructions. Briefly, cells were incubated

sequentially with different fractionation buffers followed by centrifugation with increasing gravitational force.

## Electron microscopy

Cells were harvested using accutase (Sigma), washed with PBS, pelleted by centrifugation and fixed for 45 min in 2.5% (v/v) glutaraldehyde buffered in cacodylate (pH 7.4) prior to recurrent centrifugation. The resulting cell pellet was embedded in 1% osmium tetroxide and dehydrated in a graded ethanol series, which was intermingled by an incubation step with uranyl acetate (between the 50% and 90% ethanol step) and finally, rinsed in propylene oxide. After embedding the pellets in epoxy resins, which polymerized for 16 hr at 60°C, semithin sections (0.5 µm) were cut using an ultramicrotome (Leica Ultracut UCT, Deerfield, IL, USA) with a diamond knife. Sections were stained with toluidine blue, placed on glass slides, and examined by light microscopy to select appropriate areas for ultrathin preparation. Ultrathin sections (50–70 nm) were cut using an ultramicrotome. Afterwards, sections were mounted on copper grids and contrasted with uranyl acetate for 2–3 hr at 42°C and lead citrate for 20 min at room temperature. These samples were analyzed and digitally documented using a FEI Tecnai G2 Spirit Biotwin TEM (Hillsboro, OR, USA) at an operating voltage of 120 kV.

## Chromatin immunoprecipitation (ChIP)

ChIP assays were performed as described previously (*Nayak et al., 2014*). Briefly, crosslinking of cells with 1.47% formaldehyde was stopped by addition of 125 mM glycine. Cells were lysed in ChIP buffer (150 mM NaCl; 50 mM Tris-HCl [pH 7.5], 5 mM EDTA, 0.5% NP40, 1% Triton X-100) supplemented with protease and phosphatase inhibitors. Nuclei were precipitated via centrifugation for 18,000 g for 2 min followed by sonication and chromatin isolation by centrifugation at 18,000 g for 10 min. After overnight incubation of chromatin with antibodies (5 µg), protein G dynabeads were added to capture the immunoprecipitated chromatin complex followed by several washes with ChIP lysis buffer with differing NaCl concentrations (150 mM, 500 mM, 150 mM). Reverse crosslinking and DNA isolation was performed by addition of 10% (wt/vol) Chelex-100 slurry directly to the beads and boiling for 10 min at 95°C. DNA was collected twice by centrifugation at 18,000 g for 1 min and dissolved in DNase/RNase-free water. Subsequently, DNA was analyzed by qPCR using SYBR green master mix (Thermo Fisher Scientific) with gene specific primer sets (*Supplementary file 2*).

## Microarray analysis

Total RNA from 293 T cells was isolated using the High Pure RNA isolation kit (Roche) and hybridized to an Illumina HumanHT-12 Microarray according to the protocol of the Genomics and Proteomics Core Facility (DKFZ, Heidelberg, Germany). Functional annotation analysis was performed with DAVID Bioinformatics Resources 6.8 (*Huang et al., 2009a*, *2009b*).

## Statistical analysis

Diagrams and statistical analysis were generated using GraphPad Prism 4. Data represent mean ± SEM (standard error mean) or ± standard deviation, as indicated. Statistical significance was determined with unpaired t-test, one-way ANOVA or two-way-ANOVA as necessary followed by Bonferroni post hoc test ($p < 0.05 = *$, $p < 0.01 = **$, $p < 0.001 = ***$). Correlation coefficients were calculated with Excel.

## Acknowledgements

We are grateful to JW Harper for critical access to CompPASS and M Hoffmeister and S Hölper for technical support with MS measurements. We thank N Raman, D McEwan, J Huber, C Osterburg, M Tuppi, M Kaulich and S Oess for readily sharing reagents and protocols. We are grateful to current and previous members of the Behrends lab for resources and discussions. We thank the DKFZ Genomics and Proteomics Core Facility for providing the Illumina Whole-Genome Expression Beadchips and related services. This work was supported by the Deutsche Forschungsgemeinschaft (DFG) within the framework of the Munich Cluster for Systems Neurology (EXC1010 SyNergy), the Cluster of Excellence 'Macromolecular Complexes' of the Goethe University Frankfurt (EXC115 CEF-MC), the Collaborative Research Center (CRC1177) and the project grant DI 931/3–1 as well as by the

LOEWE grant Ub-Net, the LOEWE Centrum for Gene and Cell Therapy Frankfurt and the European Research Council (ERC, 282333-XABA).

## Additional information

### Competing interests

ID: Senior Editor, *eLife*. The other authors declare that no competing interests exist.

### Funding

| Funder | Grant reference number | Author |
|---|---|---|
| Deutsche Forschungsgemeinschaft | SFB1177 | Stefan Müller<br>Ivan Dikic<br>Christian Behrends |
| LOEWE Zentrum | Ub-net | Stefan Müller<br>Ivan Dikic<br>Christian Behrends |
| Goethe-Universität Frankfurt am Main | EXC115 | Ivan Dikic |
| Deutsche Forschungsgemeinschaft | DI 931/3-1 | Ivan Dikic |
| LOEWE Zentrum | Gene and Cell Therapy Frankfurt | Ivan Dikic |
| Munich Cluster of Systems Neurology | EXC 1010 SyNergy | Christian Behrends |
| European Research Council | ERC, 282333-XABA | Christian Behrends |

The funders had no role in study design, data collection and interpretation, or the decision to submit the work for publication.

### Author contributions

JJ, Conceptualization, Data curation, Formal analysis, Investigation, Methodology, Writing—original draft, Writing—review and editing; AN, Data curation, Methodology, Writing—review and editing, Conducted ChIP experiments; VS, Data curation, Methodology, Writing—review and editing, Performed several experiments with C9ORF72 and WDR41; TS, Data curation, Methodology, Conducted EM studies; AKK, Software, Methodology, Helped developing image analysis scripts; SM, ID, Supervision, Funding acquisition, Writing—review and editing; MM, Data curation, Methodology, Writing—review and editing, Conducted EM studies; CB, Conceptualization, Supervision, Funding acquisition, Writing—original draft, Project administration, Writing—review and editing

### Author ORCIDs

Jennifer Jung, http://orcid.org/0000-0001-9436-4021
Ivan Dikic, http://orcid.org/0000-0001-8156-9511
Christian Behrends, http://orcid.org/0000-0002-9184-7607

## Additional files

### Supplementary files

• Supplementary file 1: siRNA sequences.
• Supplementary file 2: Primer sequences.

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
