## [Decision Letter]

[Editors’ note: a previous version of this study was rejected after peer review, but the authors submitted for reconsideration. The first decision letter after peer review is shown below.]

Thank you for submitting your work entitled "SMCR8 functions as negative autophagy regulator by inhibiting ULK1 kinase activity and gene expression" for consideration by *eLife*. Your article has been reviewed by three peer reviewers, one of whom is a member of our Board of Reviewing Editors, and the evaluation has been overseen by Tony Hunter as the Senior Editor. The reviewers have opted to remain anonymous.

Our decision has been reached after consultation between the reviewers. Based on these discussions and the individual reviews below, we regret to inform you that your work will not be considered further for publication in *eLife* at this time.

While the first part of the paper is interesting and informative, the second part raises at least two major concerns.

1) It is unclear how SMCR8 regulates the ULK1 kinase activity (Reviewer #1 – Point #4-6, Reviewer #2 – Point #1, Reviewer #3 – Point #1).

2) The nature of the ULK1-SMCR complex is not sufficiently characterized (Reviewer #1 – Point #1 and 4, Reviewer #2 – Point #1, Reviewer #3 – Point #2.).

The reviews contain a number of additional major concerns that also need to be dealt with.

If you were to address these major concerns, the experiments would take more than the usual two-month period that *eLife* requires to ensure speedy decisions.

However, if you can address these criticisms of the reviewers and rewrite the manuscript, we would be willing to consider a resubmission to *eLife* in the future. Otherwise, we hope that these comments would be helpful to submit other journal.

*Reviewer #1:*

Jung et al. performed an image-based RNAi screen of 186 Rabs and their regulators, and identified 34 autophagy regulators. Among them, RAB27A, RAB27B, MADD, DENND2C, RAB36, TBC1D8, and SMCR8 were further investigated and confirmed to have potential autophagy-regulating functions by electron microscopy. MADD, RAB27A, DENND2C, and SMCR8 also interact with known autophagy factors. In the second half part, the authors focus on SMCR8, which interacts with FIP200/RB1CC1, the Rab GEF C9ORF72, and its cofactor WDR41. AA 120-320 of SMCR8 interacts with ATG13, whereas a longer region (AA 1-716) is required for the interaction with ULK1 and FIP200. The authors further suggest that SMCR8 regulates not only the autophagosome maturation step (recently published by another group) but also the autophagy initiation step through upregulation of the ULK1 kinase activity and ULK1 gene expression.

This study identifies a novel regulatory mechanism of autophagy. Although SMCR8 was recently identified as a binding partner of C9ORF72 that relates to the pathogenesis of ALS/FTD, the autophagy-initiation effect of SMCR8 is likely independent of C9ORF72. Thus, the reported function of SMCR8 is novel and different from several recent studies on C9ORF72.

1) It is difficult to conceive that the fragment 120-320 of SMCR8 is sufficient for binding with ATG13 but not for ULK1 and FIP200. How much of SMCR8 is included in the large (~3 MDa) ATG13-FIP200-ULK1 complex (Hosokawa et al. Mol Biol Cell 20, 1981, 2009, Hieke et al. Autophagy 11,1471 2015)? Is the incorporation of SMCR8 into the ATG13-FIP200-ULK1 complex regulated by starvation? These points are important to propose a physiological function of SMCR8 in autophagy regulation.

2) Figure 5: In the presence of bafilomycin A1, there is no significant difference between sicon and siSMCR8. It suggests that autophagy-initiating role of SMCR8 is relatively minor, which contradicts with the statement in the text. This needs a clearer explanation.

3) The evidence of mTOR-independency is not strong. Phosphorylation of mTORC1 substrates such as S6K1 and 4E-BP1 should be checked in SMCR8 knockout or knockdown cells.

4) Figure 7: The authors show that knockdown of SMCR8 increases ULK1 kinase activity because the phosphorylation level of ATG13 is increased. However, it is also possible that SMCR8 simply inhibits the interaction between ULK1 and ATG13 without affecting the ULK1 kinase activity. Does overexpression of SMCR8 break the ULK1-ATG13 interaction? In addition, it is recommended that the ULK1 kinase activity should be determined using other substrates such Beclin 1 (Russell et al. Nat Cell Biol. 2013 15:741-50) and ATG14 (Park et al. Autophagy 2016 12:547-564).

5) Figure 7: The authors suggest that the autophagy-initiation effect of SMCR8 is independent of C9ORF72 and WDR41 only by seeing the expression level of ULK1 in siC9ORF72 and siWDR41 cells. This is not sufficient. The ULK1 activity should also be measured in these cells.

6) It is unclear whether the ULK1 kinase activity and the ULK1 expression level are independently regulated by SMCR8. The authors show that SMCR8 1-700 fails to translocate to the nucleus (Figure 8). To distinguish the regulatory role of SMCR8 in ULK1 gene expression from that in ULK1 kinase activity, the autophagic flux and ULK1 kinase activity should be determined in SMCR8 knockout cells rescued with SMCR8 1-700.

*Reviewer #2:*

In this manuscript, Jung and coworkers performed a comprehensive RNAi screening for Rabs and their regulatory enzymes that control autophagy and identified 34 potential candidates. Among them, they focused on a DENN protein SMCR8, which is known to form a complex with C9ORF72, a recently reported autophagy regulator, and showed that unlike C9ORF72 SMCR8 negatively regulates an early phase of autophagy presumably through inhibiting ULK1 kinase activity and its gene expression. Although the manuscript contains novel information that would be interested in many readers in the field, additional experiments and revisions are required to strengthen the authors' conclusions.

Specific points:

1) Because C9ORF72, a binding partner of SMCR8, can also interact with the ULK1 complex (Webster et al. (2016) EMBO J.), it is unclear whether the ULK1 complex interacts with the C9ORF72/SMCR8/WDR41 complex or the individual components (see Figure 9 model). Direct binding and competition assays are required to understand the molecular mechanism by which SMCR8 inhibits initiation of autophagy. Without these data, the authors cannot exclude the possibility that SMCR8 directly inhibits the positive function of C9ORF72 upon autophagy initiation.

2) How do the authors obtain a SMCR8 KO cell line? By CRISPR/Cas9 or TALEN? There is no description about the SMCR8 KO cells in the Materials and methods section. Furthermore, this reviewer cannot understand the reason why the authors used the KO cells only in a few experiments (Figure 5 and Figure 7). Since the KD efficiency of siSMCR8 is not so high (0.43 and 0.63 in Figure 2), most of the experiments shown in Figure 5–Figure 8 should be reinvestigated by using the KO cells and SMCR8-reexpressing KO cells (rescued cells). Furthermore, EM analyses of SMCR8 KO cells and rescued cells, including quantitative analysis, should also be performed in Figure 2 to confirm the authors' original finding.

3) In Figure 8, many immunoreactive bands were detected with anti-SMCR8 antibody. Is ~70-kDa immunoreactive band a nonspecific band or a degradation product of SMCR8? The SMCR8 KO cells should be used in Figure 8 as a negative control to show the specificity of the antibody.

4) This reviewer does not see any specific endogenous SMCR8 band in the nucleoplasm or chromatin fraction. More convincing evidence is required. In addition to ChIP assay, the authors should perform a reporter assay to determine whether SMCR8 indeed inhibits ULK1 transcription.

5) Although the authors claimed that SMCR8 inhibits ULK1 kinase activity, the amount of p-ULK1/total ULK1 seemed not to be increased in SMCR8 KD cells (Figure 7). Phosphorylation data shown in Figure 7 should be quantified (p-ULK1/total ULK1 and p-ATG13/total ATG13) and analyzed statistically.

6) C9ORF72 has recently been reported to be involved in initiation of autophagy rather than its maturation through interaction with Rab1a and the ULK1 complex (Webster et al. (2016) EMBO J.). This paper should be cited and descried in the text.

*Reviewer #3:*

In the manuscript by Jung et al., the authors performed an imaging-based RNAi screen monitoring different populations of autophagosomes in order to investigate the role of Rab GTPases and the corresponding GAPs and GEFs. The authors focused on seven candidate genes for further analysis, including ultrastructural analysis and the determination of the interactomes. In the second half of the manuscript, the authors investigated to role of the candidate protein SMCR8 for autophagy. They found that both kinase activity and gene expression of ULK1 is increased in SMCR8-depleted cells. Collectively, the authors suggest that SMCR8 is a dual negative regulator of ULK1.

Generally, the manuscript is well written and the experiments are carefully planned and conducted. This holds especially true for the first half of the manuscript describing the RNAi-based screening approach. The results of the second part of the manuscript – describing the relation between SMCR8 and ULK1 – are also very interesting and an important work for the research field of the autophagy-inducing ULK1 complex. However, prior to publication on *eLife*, I suggest that the authors address some remaining issues. These additional experiments might then strengthen the authors' conclusion.

My comments are as follows:

Major points:

1) The authors state that SMCR8 negatively affects ULK1 kinase activity. As readout, the authors make use of the detection of ATG13 phosphorylation at Ser318. I agree that this is a suitable readout, but I recommend the inclusion of additional approaches, e.g. in vitro kinase assays with purified ULK1 from mock- or SMCR8-siRNA-treated cells. Although I agree that the increased ATG13 phosphorylation is not necessarily caused by the increased ULK1 levels in SMCR8-depleted cells (nicely confirmed in Figure 7), there exist alternative explanations for the increased ATG13 phosphorylation (altered binding stability between ATG13 and ULK1, or steric hindrance in the presence of SMCR8).

2) The authors investigate the interaction between SMCR8 and the ULK1 complex. I think there are several issues with this point:

a) Some of the co-purifications shown in Figure 4 are not consistent between panel E and F (e.g. ULK1 association with fragment 1-270, or FIP200 association with 271-700). Obviously different exposures are shown. Can this be adjusted? Or normalized to the used HA- SMCR8 fragments?

b) I think it would be nice to know which component of the ULK1 complex mediates binding to SMCR8. The KO cells for the single components are available, and accordingly the authors should investigate this aspect. Alternatively, recombinant proteins could be employed.

c) In the Discussion section the authors suggest that SMCR8 regulates autophagy through interaction with two distinct complexes, i.e. the ULK1 complex and C9ORF72. This is obviously a very intriguing hypothesis. One potential experiment to investigate this would be size exclusion experiments. Alternatively, it could be investigated whether C9ORF2 can be co-purified with immunopurifications of components of the ULK1 complex or vice versa. Along these lines, Webster et al. have recently reported that C9ORF72 IPs contain the components of the ULK1 complex (PMID 27334615), indicating that there are not necessarily two disctinct SMCR8 complexes.

[Editors’ note: what now follows is the decision letter after the authors submitted for further consideration.]

Thank you for submitting your article "SMCR8 functions as negative autophagy regulator by inhibiting ULK1 kinase activity and gene expression" for consideration by *eLife*. Your article has been reviewed by three peer reviewers, one of whom is a member of our Board of Reviewing Editors, and the evaluation has been overseen by Tony Hunter as the Senior Editor. The reviewers have opted to remain anonymous.

The reviewers have discussed the reviews with one another and the Reviewing Editor has drafted this decision to help you prepare a revised submission.

Summary:

The authors have addressed the previous criticisms and performed several of the suggested experiments to address critical points. However, while the first part is extensive and informative, the second part describing the analyses of the SCMR8-mediated effects on ULK1 activity and expression remains somewhat premature. The following points need to be addressed.

Essential revisions:

1) In this revised version, the authors suggest the existence of a trimeric C9ORF72-WDR41-SMCR8 complex and a C9ORF72-WDR41-SMCR8/ULK1 complex holo-assembly. However, some important mechanistic details remain elusive:

a) The authors state that the interaction between SMCR8/C9ORF72 and FIP200 increases upon autophagy induction (Figure 4). These data are important but not in depth. To fully support the authors' model, it is important to perform their size exclusion experiments shown in Figure 6 also under pro-autophagic conditions and compare them to full medium conditions.

b) Figure 6 is described as follows: "…confirmed the distribution of SMCR8 in a C9ORF72-WDR41 complex and a C9ORF72-WDR41-ULK1 complex assembly", but this is not evident. Which band represents SMCR8? In several immunoblots shown in the manuscript, SMCR8 siRNA affects only the lower band. It is essential to indicate the true SMCR8 band. Size exclusion experiments with SMCR8-knockdown cell lysates would also be informative.

c) In Figure 5, the authors analyze whether SMCR8 siRNA affects interactions within the ULK1 complex. Quantification and normalization of co-purified proteins to IPed proteins (also for Figure 5 for example) is required. It appears that reduced amounts of ATG13 are purified with ULK1 upon SMCR8 depletion.

2) The lack of phenotype in SMCR8-depleted cells that have been cultured for a long period (> 4 weeks) is important information for the general readers. The authors should describe this fact in the main text, and include Figure 12 (compensatory mechanism after SMCR8 depletion) in the supplemental information.

3) In the legend to Figure 9, at least 3 experiments should be performed for statistical analysis (currently n = 2!).

---

## [Author Response]

[Editors’ note: a previous version of this study was rejected after peer review, but the authors submitted for reconsideration. The first decision letter after peer review is shown below.]

*Reviewer #1:*

*[…] 1) It is difficult to conceive that the fragment 120-320 of SMCR8 is sufficient for binding with ATG13 but not for ULK1 and FIP200. How much of SMCR8 is included in the large (~3 MDa) ATG13-FIP200-ULK1 complex (Hosokawa et al. Mol Biol Cell 20, 1981, 2009, Hieke et al. Autophagy 11,1471 2015)? Is the incorporation of SMCR8 into the ATG13-FIP200-ULK1 complex regulated by starvation? These points are important to propose a physiological function of SMCR8 in autophagy regulation.*

To address the reviewer’s concerns, we quantified our HA-IP immunoblots with SMCR8 truncation mutants (rearranged Figure 5 as well as rearranged Figure 5—figure supplement 1), which confirmed that the SMCR8 fragment 120-320 is necessary and sufficient to associate with ATG13, while ULK1 and FIP200 require a longer SMCR8 fragment for binding. This indicates that SMCR8 binding to the ULK1 complex is mediated by association with more than one subunit. Moreover, we examined the complex composition of SMCR8 and ULK1 assemblies via a series of IP experiments followed by Native PAGE or size exclusion chromatography and mass spectrometry (MS) analysis (new Figure 6). In addition, we analyzed cell lysates by size exclusion chromatography and immunoblotting (new Figure 6). In summary, these experiments point to the co-existence of a separate SMCR8-C9ORF72-WDR41 complex and a SMCR8-C9ORF72-WDR41-ULK1 complex holo-assembly. Finally, we examined the association of SMCR8 and C9ORF72 with ULK1 complex components during autophagy induction by Torin1 or starvation (new Figure 4). Finally, association of FIP200 with SMCR8 and C9ORF72 markedly increased upon autophagy induction, while binding of SMCR8 to ATG13 decreased, indicating that binding of SMCR8 to C9ORF72 and the ULK1 complex is differentially regulated.

*2) Figure 5: In the presence of bafilomycin A1, there is no significant difference between sicon and siSMCR8. It suggests that autophagy-initiating role of SMCR8 is relatively minor, which contradicts with the statement in the text. This needs a clearer explanation.*

We agree that increased LC3-II levels upon knockdown of SMCR8 are barely visible in our initial Figure 5. To provide a stronger LC3B phenotype we focused on HAP1 SMCR8 knockout cells. However, SMCR8 knockout cells seemingly developed a compensatory mechanism to potentially cope with SMCR8 deletion, which results in a lack of the initially detected phenotypes after a few passages and freeze-thaw cycles (Figure 12). Since a recent paper (Yang et al. 2016) nicely elaborates on the effect of ablated SMCR8 on LC3B, we removed our LC3B immunoblot (former Figure 5) from the manuscript, moved the autophagy figures into the figure supplements (rearranged Figure 6—figure supplement 1 and Figure 6—figure supplement 2) and focused our efforts on novel insights on SMCR8’s role in regulating transcription of ULK1, WIPI2 and other autophagy- related genes (revised Figure 8, revised Figure 9 and new Figure 10).

Author response image 1.(A-B) Early (**A**) or late (**B**) passages of HAP1 SMCR8 wildtype (wt) or knockout (ko) cells were subjected to SDS-PAGE and immunoblot with indicated antibodies.PCNA serves as loading control. (**C**) 293T SMCR8 wildtype (wt) or knockout (ko) cells were analyzed as in (**A**). (**D**) 293T cells were transfected with non-targeting control (sicon) or SMCR8 siRNA. Half of the cells were re-transfected every 2-3 days with non-targeting control (sicon) or SMCR8 siRNA while the other half was harvested and subjected to SDS-PAGE and immunoblotting with indicated antibodies. PCNA served as loading control.**DOI:**
http://dx.doi.org/10.7554/eLife.23063.036

*3) The evidence of mTOR-independency is not strong. Phosphorylation of mTORC1 substrates such as S6K1 and 4E-BP1 should be checked in SMCR8 knockout or knockdown cells.*

As requested we tested phosphorylation of the mTORC1 substrates S6K1 and 4EBP1 upon SMCR8 depletion and quantified our phospho-ULK1 immunoblots (revised Figure 7). Intriguingly, SMCR8 knockdown induced an increase in ULK1 and S6K phosphorylation but also elevated both protein levels. Ratio determination of phosphorylated and total protein levels reveals a slight reduction of mTORC1 dependent phosphorylation on these substrates. However, the mTORC1 substrate 4EBP1 remained largely unchanged in respect to phosphorylation or protein levels. These data suggest only a minor effect of SMCR8 on mTORC1.

*4) Figure 7: The authors show that knockdown of SMCR8 increases ULK1 kinase activity because the phosphorylation level of ATG13 is increased. However, it is also possible that SMCR8 simply inhibits the interaction between ULK1 and ATG13 without affecting the ULK1 kinase activity. Does overexpression of SMCR8 break the ULK1-ATG13 interaction? In addition, it is recommended that the ULK1 kinase activity should be determined using other substrates such Beclin 1 (Russell et al. Nat Cell Biol. 2013 15:741-50) and ATG14 (Park et al. Autophagy 2016 12:547-564).*

We thank the reviewer for this comment. To strengthen our hypothesis that lack of SMCR8 caused enhanced ULK1 kinase activity, we monitored phosphorylation of ATG14 on S29 and BECN1 on S15 as two other suggested ULK1 substrates. Consistent with our results on ATG13, depletion of SMCR8 led to increased levels of phospho-ATG14 S29 (new Figure 7). Unfortunately, PPase treatment for 1 h left the phosphorylation status of Beclin1 unchanged (p-BECN1 antibody, Abbiotec #254515) (Figure 13) Therefore, we did not include Beclin-1 in our analysis of SMCR8 dependent ULK1 regulation. Furthermore, neither SMCR8 overexpression nor SMCR8 depletion induced changes in the ULK1 complex composition as revealed by IP with antibodies against endogenous ULK1 and FIP200 (new Figure 5). In conclusion, these experiments provide strong evidence for regulation of ULK1 kinase activity by SMCR8 although the mechanism remains undefined.

Author response image 2.Evaluation of phospho-antibody specificity.(**A**) Lysates from 293T cells transfected with HA-tagged Beclin1 were subjected to HA-IP followed by Lambda Protein Phosphatase (PPase) treatment for 1 h and SDS-PAGE and immunoblotting with indicated antibodies.**DOI:**
http://dx.doi.org/10.7554/eLife.23063.037

*5) Figure 7: The authors suggest that the autophagy-initiation effect of SMCR8 is independent of C9ORF72 and WDR41 only by seeing the expression level of ULK1 in siC9ORF72 and siWDR41 cells. This is not sufficient. The ULK1 activity should also be measured in these cells.*

As requested, we tested ULK1-dependent ATG13 phosphorylation in C9ORF72 or WDR41 depleted cells (new Figure 7). In contrast to cells depleted of SMCR8, knockdown of C9ORF72 and WDR41 unexpectedly decreased ATG13 phosphorylation, indicating an involvement of C9ORF72 in autophagy initiation. However, ULK1 protein abundance was unimpaired in C9ORF72 depleted cells (rearranged Figure 8), suggesting a GEF complex independent role of SMCR8 on gene expression, which we focused our further efforts on (revised Figure 8, revised Figure 9 and new Figure 10).

*6) It is unclear whether the ULK1 kinase activity and the ULK1 expression level are independently regulated by SMCR8. The authors show that SMCR8 1-700 fails to translocate to the nucleus (Figure 8). To distinguish the regulatory role of SMCR8 in ULK1 gene expression from that in ULK1 kinase activity, the autophagic flux and ULK1 kinase activity should be determined in SMCR8 knockout cells rescued with SMCR8 1-700.*

We absolutely agree that dissecting the different contributions of SMCR8 to autophagy initiation, autophagosome maturation and autophagy gene expression regulation requires more effort. Our trials to re-express SMCR8 full-length and fragments in SMCR8 knockout cells were unsuccessful, since knockout cells seemingly develop a compensatory mechanism to antagonize the lack of SMCR8 (Figure 12). Therefore, we replaced our former knockout rescue experiment (former Figure 7) with an experiment using re-expression of SMCR8 full-length protein after siRNA-mediated SMCR8 knockdown. In this rescue condition, ULK1 protein levels are comparable to control cells (new Figure 8). Similar complementation experiments with SMCR8 fragments would of course be very informative but exceeded our time limits in the light of substantial competition with several other research groups (Amick, Roczniak-Ferguson, and Ferguson 2016; Sellier et al. 2016; Sullivan et al. 2016; Yang et al. 2016).

*Reviewer #2:*

*In this manuscript, Jung and coworkers performed a comprehensive RNAi screening for Rabs and their regulatory enzymes that control autophagy and identified 34 potential candidates. Among them, they focused on a DENN protein SMCR8, which is known to form a complex with C9ORF72, a recently reported autophagy regulator, and showed that unlike C9ORF72 SMCR8 negatively regulates an early phase of autophagy presumably through inhibiting ULK1 kinase activity and its gene expression. Although the manuscript contains novel information that would be interested in many readers in the field, additional experiments and revisions are required to strengthen the authors' conclusions.*

*Specific points:*

*1) Because C9ORF72, a binding partner of SMCR8, can also interact with the ULK1 complex (Webster et al. (2016) EMBO J.), it is unclear whether the ULK1 complex interacts with the C9ORF72/SMCR8/WDR41 complex or the individual components (see Figure 9 model). Direct binding and competition assays are required to understand the molecular mechanism by which SMCR8 inhibits initiation of autophagy. Without these data, the authors cannot exclude the possibility that SMCR8 directly inhibits the positive function of C9ORF72 upon autophagy initiation.*

We thank the reviewer for this comment, which prompted us to probe the complex composition of the SMCR8-C9ORF72 and ULK1 assemblies with a series of IP experiments. Intriguingly, binding studies of SMCR8 and C9ORF72 with the ULK1 complex components revealed almost no association of the ULK1 complex with C9ORF72 in comparison to SMCR8 under basal condition. However, interaction of C9ORF72 and SMCR8 with FIP200 markedly increased upon autophagy induction (new Figure 4). Furthermore, we tested whether overexpression of GFP-ATG13 or GFP- C9ORF72 competes with other SMCR8 associated proteins for binding to SMCR8. Neither ATG13 nor C9ORF72 was able to outcompete C9ORF72 or ATG13, respectively, from SMCR8 immune complexes (new Figure 5). Finally, we applied Native PAGE or size exclusion chromatography followed by MS analysis or immunoblotting to study the composition of the complex assembly (new Figure 6). In summary, these experiments point to co-existence of a separate SMCR8-C9ORF72-WDR41 complex and a SMCR8-C9ORF72-WDR41-ULK1 complex holo-assembly. We agree that in vitro binding assays with purified proteins would still provide an even deeper insight into the exact complex composition. However, we feel that establishing protein purification protocols for seven proteins is an enormous task and beyond the scope of this manuscript. Furthermore, despite the common function of SMCR8, C9ORF72 and WDR41, SMCR8 clearly has an independent role in transcriptional regulation of autophagy genes, since C9ORF72 and WDR41 depletion left ULK1 protein levels unchanged (rearranged Figure 8).

*2) How do the authors obtain a SMCR8 KO cell line? By CRISPR/Cas9 or TALEN? There is no description about the SMCR8 KO cells in the Materials and methods section.*

We are very sorry for this omission and included following sentence into Materials and methods section of the manuscript: HAP1 SMCR8 knockout cells were purchased from Horizon Discovery (HZGHC003606c011).

*Furthermore, this reviewer cannot understand the reason why the authors used the KO cells only in a few experiments (Figure 5 and Figure 7). Since the KD efficiency of siSMCR8 is not so high (0.43 and 0.63 in Figure 2), most of the experiments shown in Figure 5–Figure 8 should be reinvestigated by using the KO cells and SMCR8-reexpressing KO cells (rescued cells).*

We thank the reviewer for this suggestion. During our effort to repeat several of the requested experiments in HAP1 SMCR8 knockout cells, we observed that these cells developed a compensatory mechanism for SMCR8 deletion and hence were devoid of the initially observed phenotype (Figure 12). Subsequently, we generated a 293T SMCR8 knockout cell line using CRISPR-Cas9. Since generation and subsequent clonal selection of these knockout cells allowed us to perform the first immunoblot only after about 6 weeks, we again were unable to detect any ULK1 phenotype (Figure 12). To investigate a potential compensatory mechanism, we performed a long-term siRNA knockdown experiment: First, we transfected 293T cells with non- targeting or SMCR8 siRNA. After 2-3 days half of the cells were re-transfected with siRNA while the other half was harvested for SDS-PAGE and immunoblotting. The same procedure was applied for several weeks. In this time course, we identified a lack of elevated ULK1 protein levels after 4 weeks of siRNA knockdown, while SMCR8 was still depleted (Figure 12). In addition, we demonstrated on-target effects of the SMCR8 siRNA by rescuing SMCR8 induced increase in ULK1 protein levels by re-expression of SMCR8 in SMCR8 knockdown cells (new Figure 8). Hence, we stopped any experiments with knockout cells and performed further experiments with siRNA mediated depletion of SMCR8.

*Furthermore, EM analyses of SMCR8 KO cells and rescued cells, including quantitative analysis, should also be performed in Figure 2 to confirm the authors' original finding.*

As requested, we quantified the number of lysosomes in 20 control and SMCR8 knockdown cells (new Figure 3—figure supplement 1). Number of lysosomes was increased upon SMCR8 depletion compared to control cells.

*3) In Figure 8, many immunoreactive bands were detected with anti-SMCR8 antibody. Is ~70-kDa immunoreactive band a nonspecific band or a degradation product of SMCR8? The SMCR8 KO cells should be used in Figure 8 as a negative control to show the specificity of the antibody.*

We included an immunoblot with parental and SMCR8 knockout HAP1 cells to show the specificity of our SMCR8 antibody (new Figure 9—figure supplement 1). Indeed, the SMCR8 antibody detects a couple of bands, three of which are diminished in SMCR8 knockout cells (~135kDa, ~100kDa, ~48 kDa) and hence specific for SMCR8. These bands are highlighted by arrows. The different bands might represent different isoforms, post-translational modified variants, cleavage or degradation products of SMCR8, which has a nominal molecular weight of 105 kDa.

*4) This reviewer does not see any specific endogenous SMCR8 band in the nucleoplasm or chromatin fraction. More convincing evidence is required. In addition to ChIP assay, the authors should perform a reporter assay to determine whether SMCR8 indeed inhibits ULK1 transcription.*

To improve our detection of endogenous SMCR8 in the nucleus, we used CRISPR- Cas9 technology to generate an endogenously tagged SMCR8 cell line (new Figure 9—figure supplement 1), which we subsequently subjected to subcellular fractionation. As detected for exogenous expressed full-length SMCR8 (rearranged Figure 9), HA- tagged endogenous SMCR8 was mainly localized to the cytoplasmic and membrane fraction (new Figure 9). However, longer exposure also revealed nuclear and even chromatin localization of endogenous SMCR8 (new Figure 9). We performed ChIP at endogenous level, which is commonly considered the gold standard for chromatin association of proteins. A reporter assay would be more artificial than the ChIP experiment. However, as an additional control, we extended the ChIP analysis at endogenous levels to FIP200 using specific primers but were unable to detect chromatin association of SMCR8 to the FIP200 gene locus (new Figure 9). These data are consistent with the immunoblot analysis of these two proteins (new Figure 8).

*5) Although the authors claimed that SMCR8 inhibits ULK1 kinase activity, the amount of p-ULK1/total ULK1 seemed not to be increased in SMCR8 KD cells (Figure 7). Phosphorylation data shown in Figure 7 should be quantified (p-ULK1/total ULK1 and p-ATG13/total ATG13) and analyzed statistically.*

As requested we quantified the ratio of the ULK1 dependent phosphorylation on ATG13 in SMCR8 depleted cells and identified a 3-fold increase compared to control cells (rearranged Figure 7). In addition, we also detected increased phosphorylation of ATG14, which represents another ULK1 substrate (new Figure 7), further supporting our findings that SMCR8 regulates ULK1 kinase activity. Since this activity is in turn regulated by phosphorylation through upstream kinases, we also quantified mTORC1 dependent phosphorylation on S757 of ULK1 upon SMCR8 knockdown. The ratio for p-ULK1(S757)/ULK1 is marginally decreased after SMCR8 depletion, which was confirmed for S6K another mTORC1 substrate. For both, increase in their phosphorylation status was accompanied with elevated protein levels. However, phosphorylation status and protein levels of another mTORC1 substrate, 4EBP1, remained unchanged. Thus, we concluded that SMCR8 only slightly effects mTORC1 activity and hence regulates ULK1 activity by a yet undiscovered mechanism.

*6) C9ORF72 has recently been reported to be involved in initiation of autophagy rather than its maturation through interaction with Rab1a and the ULK1 complex (Webster et al. (2016) EMBO J.). This paper should be cited and descried in the text.*

We apologize for this omission. The paper was not published yet when we initially submitted our manuscript to *eLife*. We now included their findings into the revised manuscript.

*Reviewer #3:*

*[…] Major points:*

*1) The authors state that SMCR8 negatively affects ULK1 kinase activity. As readout, the authors make use of the detection of ATG13 phosphorylation at Ser318. I agree that this is a suitable readout, but I recommend the inclusion of additional approaches, e.g.* in vitro *kinase assays with purified ULK1 from mock- or SMCR8-siRNA-treated cells. Although I agree that the increased ATG13 phosphorylation is not necessarily caused by the increased ULK1 levels in SMCR8-depleted cells (nicely confirmed in Figure 7), there exist alternative explanations for the increased ATG13 phosphorylation (altered binding stability between ATG13 and ULK1, or steric hindrance in the presence of SMCR8).*

We thank the reviewer for this advice. To further establish a role for SMCR8 in regulating ULK1 kinase activity, we confirmed the SMCR8-dependent increase in phosphorylation of ATG14 as another ULK1 substrate (new Figure 7). Unfortunately, PPase treatment for 1 h left the phosphorylation status of Beclin1 unchanged (p- BECN1 antibody, Abbiotec #254515) (Figure 13). Therefore, we did not include Beclin-1 in our efforts to monitor SMCR8 dependent ULK1 regulation. Furthermore, we probed the ULK1 complex composition at endogenous levels after SMCR8 overexpression or knockdown and observed no change in association of ULK1 with ATG13 or FIP200 (new Figure 5). Taken together, these data suggest a modulation of ULK1 kinase activity through a yet to be defined mechanism.

*2) The authors investigate the interaction between SMCR8 and the ULK1 complex. I think there are several issues with this point:*

*a) Some of the co-purifications shown in Figure 4 are not consistent between panel E and F (e.g. ULK1 association with fragment 1-270, or FIP200 association with 271-700). Obviously different exposures are shown. Can this be adjusted? Or normalized to the used HA- SMCR8 fragments?*

As recommended we replaced the blots with more equal exposures (rearranged Figure 5 as well as rearranged Figure 5—figure supplement 1). Furthermore, we quantified and normalized the co-immunoprecipitated proteins to the amount of IPed HA-SMCR8 fragments (new Figure 5 as well as Figure 5—figure supplement 1). While endogenous FIP200 can be co-immunoprecipitated with several SMCR8 fragments (e.g. 1-500; 120-700) and SMCR8 full length protein, the greatest amount is co- immunoprecipitated with the SMCR8 fragment 1-700. The same is true for ULK1 and consistent throughout both fragment HA-IP figures.

*b) I think it would be nice to know which component of the ULK1 complex mediates binding to SMCR8. The KO cells for the single components are available, and accordingly the authors should investigate this aspect. Alternatively, recombinant proteins could be employed.*

We would like to point out that employing single ULK1 complex component knockout cells to elucidate the exact complex composition of the ULK1-SMCR8 assembly has technical limitations since ULK1 complex components stabilize each other. For example, knockdown of ATG13 leads to decreased protein abundance of ULK1 and FIP200 (Hosokawa et al. 2009). We agree that in vitro binding assays with purified proteins would provide an even deeper insight into the exact complex composition. However, we feel that establishing protein purification protocols for seven proteins is an enormous task and beyond the scope of this manuscript. As alternative approach, we applied a series of IP experiments. Neither overexpressed GFP-ATG13 nor GFP-C9ORF72 were able to outcompete C9ORF72 or ATG13, respectively, from SMCR8 immune complexes (new Figure 5). In addition, SMCR8 overexpression or knockdown left ULK1 complex composition unimpaired, as tested with immunoprecipitation using anti-ULK1 and anti-FIP200 antibodies at endogenous levels (new Figure 5).

*c) In the Discussion section the authors suggest that SMCR8 regulates autophagy through interaction with two distinct complexes, i.e. the ULK1 complex and C9ORF72. This is obviously a very intriguing hypothesis. One potential experiment to investigate this would be size exclusion experiments. Alternatively, it could be investigated whether C9ORF2 can be co-purified with immunopurifications of components of the ULK1 complex or vice versa. Along these lines, Webster et al. have recently reported that C9ORF72 IPs contain the components of the ULK1 complex (PMID 27334615), indicating that there are not necessarily two disctinct SMCR8 complexes.*

To address these reviewers’ points, we performed IP experiments followed by Native PAGE or size exclusion chromatography and MS analysis as well as size exclusion chromatography and immunoblotting on cell lysates to study the composition of the complex assembly (new Figure 6). In summary, these experiments point to co-existence of a separate SMCR8-C9ORF72-WDR41 complex and a SMCR8-C9ORF72-WDR41-ULK1 complex holo-assembly. Parallel HA-IP of HA-tagged SMCR8 and C9ORF72 revealed almost no binding to C9ORF72 with the ULK1 complex components in comparison to SMCR8 under basal conditions. However, interaction of C9ORF72 and SMCR8 to FIP200 markedly increased after autophagy induction by starvation (new Figure 4). Thus, we postulate an important regulative function of SMCR8 and C9ORF72 within the SMCR8-C9ORF72-WDR41-ULK1 complex holo-assembly.

[Editors’ note: what now follows is the decision letter after the authors submitted for further consideration.]

*Essential revisions:*

*1) In this revised version, the authors suggest the existence of a trimeric C9ORF72-WDR41-SMCR8 complex and a C9ORF72-WDR41-SMCR8/ULK1 complex holo-assembly. However, some important mechanistic details remain elusive:*

*a) The authors state that the interaction between SMCR8/C9ORF72 and FIP200 increases upon autophagy induction (Figure 4). These data are important but not in depth. To fully support the authors' model, it is important to perform their size exclusion experiments shown in Figure 6/C also under pro-autophagic conditions and compare them to full medium conditions.*

To address the SMCR8-C9ORF72-WDR41-ULK1 complex holo-assembly under autophagy induction, we treated empty or HA-ATG13 overexpressing 293T cells with DMSO or Torin1 and performed size exclusion chromatography and immunoblot or MS analysis on whole cell lysates or eluted ATG13 immune complexes. We could not detect major differences in the distribution of SMCR8, C9ORF72, WDR41 or the ULK1 complex components when comparing Torin1 treated and untreated conditions (new Figure 6—figure supplement 1). Notably, despite equal input levels the amount of ATG13 immunoprecipitated in Torin1 treated cells is 1.6 fold higher than in controls cells. This increase is also reflected in the co-immunoprecipitated proteins. Size exclusion chromatography might lack the resolution and sensitivity to accurately and quantitatively detect the dynamics of the SMCR8-C9ORF72-WDR41-ULK1 complex holo-assembly. More sophisticated approaches such as cross-linking MS or cryo-EM would be required to acquire a low-resolution structural view of the holo-assembly. However, we feel that application of these techniques is beyond the scope of this manuscript. The enhanced interaction of SMCR8 and C9ORF72 with FIP200 (Figure 4) still suggests that the SMCR8-C9ORF72-WDR41-ULK1 complex holo-assembly is preferentially formed upon autophagy induction.

*b) Figure 6 is described as follows: "…confirmed the distribution of SMCR8 in a C9ORF72-WDR41 complex and a C9ORF72-WDR41-ULK1 complex assembly", but this is not evident. Which band represents SMCR8? In several immunoblots shown in the manuscript, SMCR8 siRNA affects only the lower band. It is essential to indicate the true SMCR8 band. Size exclusion experiments with SMCR8-knockdown cell lysates would also be informative.*

To unequivocally establish the specificity of the SMCR8 antibody, we performed the proposed experiment and transiently transfected 293T cells with control or SMCR8 siRNA, followed by cell lysis and size exclusion chromatography (new Figure 6—figure supplement 1). The immunoblot with anti-SMCR8 antibody demonstrated reduced intensity of the band at 135 kDa in SMCR8 knockdown compared to control cells. The mentioned upper band in total cell lysate immunoblots is not visible after size exclusion chromatography. We further extended the description of these experiments in the manuscript and indicated the SMCR8 complexes with bars in revised Figure 6.

*c) In Figure 5, the authors analyze whether SMCR8 siRNA affects interactions within the ULK1 complex. Quantification and normalization of co-purified proteins to IPed proteins (also for Figure 5 for example) is required. It appears that reduced amounts of ATG13 are purified with ULK1 upon SMCR8 depletion.*

We quantified and normalized three independent experiments for Figure 5 and included the results as heatmap below the immunoblots (revised Figure 5). The interaction between SMCR8 and C9ORF72 or ULK1 complex components remain unimpaired upon ATG13 or C9ORF72 overexpression. Furthermore, we substituted former Figure 5 showing endogenous IP using SMCR8 knockdown cells with SMCR8 knockout cells and again quantified and normalized three independent experiments (revised Figure 5). Since knockdown of SMCR8 caused an increase in ULK1 protein levels, the exact analysis of the amount of co-immunoprecipitated ULK1 complex members was imprecise. In consequence, we performed endogenous immunoprecipitations with SMCR8 knockout cells, in which ULK1 protein levels are constant. Immunoprecipitation of ULK1 and FIP200 at endogenous levels retrieved equal amounts of each other and ATG13 after SMCR8 deletion. Notably, size exclusion chromatography with SMCR8 knockdown cells demonstrated unaltered ULK1 complex composition compared to control cells. These data provide strong evidence that the ULK1 complex composition is unimpaired in absence of SMCR8.

*2) The lack of phenotype in SMCR8-depleted cells that have been cultured for a long period (> 4 weeks) is important information for the general readers. The authors should describe this fact in the main text, and include Figure 12 (compensatory mechanism after SMCR8 depletion) in the supplemental information.*

As recommended, we included the figure in the manuscript as Figure 8—figure supplement 1 and described the findings in the manuscript.

*3) In the legend to Figure 9, at least 3 experiments should be performed for statistical analysis (currently n = 2!).*

We repeated the ChIP experiment in Figure 9 a third time. Improved statistics determined that the chromatin enrichment of SMCR8 on the ULK1 gene locus is significant, while SMCR8 knockdown reduces this enrichment significantly.